# LIS1 determines cleavage plane positioning by regulating actomyosin-mediated cell membrane contractility

**Hyang Mi Moon[1][†]\*, Simon Hippenmeyer[2][‡], Liqun Luo[2], Anthony Wynshaw-Boris[1,3]\***

[1]Department of Pediatrics, Institute for Human Genetics, Eli and Edythe Broad Center of Regenerative Medicine and Stem Cell Research, University of California, San Francisco, San Francisco, United States; [2]Howard Hughes Medical Institute and Department of Biology, Stanford University, Stanford, United States; [3]Department of Genetics and Genome Sciences, Case Western Reserve University, School of Medicine, Cleveland, United States

**Abstract** Heterozygous loss of human *PAFAH1B1* (coding for LIS1) results in the disruption of neurogenesis and neuronal migration via dysregulation of microtubule (MT) stability and dynein motor function/localization that alters mitotic spindle orientation, chromosomal segregation, and nuclear migration. Recently, human- induced pluripotent stem cell (iPSC) models revealed an important role for LIS1 in controlling the length of terminal cell divisions of outer radial glial (oRG) progenitors, suggesting cellular functions of LIS1 in regulating neural progenitor cell (NPC) daughter cell separation. Here, we examined the late mitotic stages NPCs in vivo and mouse embryonic fibroblasts (MEFs) in vitro from *Pafah1b1*-deficient mutants. *Pafah1b1*-deficient neocortical NPCs and MEFs similarly exhibited cleavage plane displacement with mislocalization of furrow-associated markers, associated with actomyosin dysfunction and cell membrane hyper-contractility. Thus, it suggests LIS1 acts as a key molecular link connecting MTs/dynein and actomyosin, ensuring that cell membrane contractility is tightly controlled to execute proper daughter cell separation.

**\*For correspondence:**
chorong@stanford.edu (HMM);
ajw168@case.edu (AW-B)

**Present address:** [†]Department of Neurosurgery, Institute for Stem Cell Biology and Regenerative Medicine, Stanford University, Stanford, United States; [‡]IST Austria (Institute of Science and Technology Austria), Klosterneuburg, Austria

**Competing interests:** The authors declare that no competing interests exist.

## Introduction

Lissencephaly (smooth brain) is a brain malformation disorder associated with haploinsufficiency of Lissencephaly-1 (LIS1), also called PAFAH1B1 (Platelet-activating factor acetylhydrolase IB subunit alpha) (*Dobyns et al., 1993*; *Hattori et al., 1994*; *Reiner et al., 1993*; *Moon and Wynshaw-Boris, 2013*). At the molecular level, LIS1 is an important regulatoratory protein controlling cytoplasmic dynein and microtubules (MTs), that is also known to have dual roles coordinating both MTs and the actin cytoskeleton (*Kholmanskikh et al., 2003*; *Kholmanskikh et al., 2006*; *Jheng et al., 2018*). Besides pivotal roles of LIS1 in post-mitotic neurononal migration, *Pafah1b1* mouse mutant studies suggest additional cellular functions of LIS1 in neocortical neural progenitor cell (NPC) division by regulating mitotic spindle orientation and cell fate (*Yingling et al., 2008*; *Youn et al., 2009*; *Hippenmeyer et al., 2010*; *Bershteyn et al., 2017*; *Moon et al., 2014*). The mitotic phenotypes of *Pafah1b1* mutants are closely related and consistent with those of other mutants of MT/dynein-associated proteins such as LGN, NDE1, and NDEL1 (*Bradshaw and Hayashi, 2017*; *Doobin et al., 2016*; *Wynne and Vallee, 2018*). However, unlike these other mouse mutants of LIS1-interacting proteins, *Pafah1b1* mutants displayed a significant decrease in neuroepithelial stem cells in the neocortex and subsequent neonatal death compared with a less catastrophic phenotype seen in radial glial (RG) progenitors (*Yingling et al., 2008*). Our recent studies with human-induced pluripotent

stem cells (iPSCs) of Miller-Dieker syndrome, a severe form of lissencephaly caused by heterozygosity of more than 20 genes including *PAFAH1B1*, demonstrated a prolongation of mitotic division time of oRG progenitors but not ventricular zone radial glial (vRG) progenitors (*Bershteyn et al., 2017*). All of these previous studies suggest that cellular functions of LIS1 other than MT/dynein-mediated mitotic spindle regulation dictate further mitotic progression from mitotic (M) phase to anaphase/telophase in late stages of mitosis and daughter cell separation.

Although the functions of LIS1 in F-actin regulation have been studied previously, its downstream cellular signaling pathways that regulate actomyosin during late mitosis and cytokinesis has not been explored. When cytokinesis is initiated, MTs provide positional cues for the cleavage furrow location and membrane ingression at the equatorial cortex by inducing central spindle assembly (*Bement et al., 2005*; *von Dassow, 2009*; *Werner et al., 2007*). At this point, dynamic crosstalk between astral MTs and cortical filamentous actin (F-actin) transmit inhibitory signals at the polar cortex, which indirectly instructs cleavage furrow positioning to the equatorial cortex (*Canman et al., 2003*; *Foe and von Dassow, 2008*; *Murthy and Wadsworth, 2008*). During cleavage plane formation, a small GTPase RhoA functions as a master regulator of cytokinetic furrow ingression (*Bement et al., 2005*; *Nishimura and Yonemura, 2006*) that recruits contractile ring components, including the scaffold protein Anillin (*D'Avino et al., 2008*; *Gregory et al., 2008*; *Oegema et al., 2000*; *Piekny and Glotzer, 2008*) and Septin (*Kinoshita et al., 2002*; *Maddox et al., 2007*; *Oegema et al., 2000*). At the same time, Myosin II accumulates at the equatorial cortex and serves as a constriction force generator by interacting with F-actin (*Glotzer, 2005*; *Straight et al., 2003*; *Zhou and Wang, 2008*). However, when coordination between MTs and F-actin is perturbed, abnormally fewer or elongated astral MTs destabilize the cleavage furrow. This MT dysfunction induces actomyosin dysregulation, leading to cell membrane hyper-contractility, RhoA and F-actin mislocalization outside of the midzone/cleavage furrow, cell shape oscillation, and subsequent mitotic failure (*Canman et al., 2003*; *Murthy and Wadsworth, 2008*; *Rankin and Wordeman, 2010*; *Watanabe et al., 2008*). It is important to note that LIS1 is involved in astral MT movements and F-actin polymerization, so it is possible that LIS1 contributes to important cellular processes during cytokinesis by regulating actomyosin and other cleavage furrow-associated proteins during late stages of mitosis and cytokinesis.

We hypothesized that the LIS1-dynein-MT network may be essential for cytokinesis, perhaps by fine-tuning actomyosin and cell membrane contractility. Therefore, we assessed the mitotic phenotypes of neocortical from *Pafah1b1*-deficient mouse embryos, and monitored the entire duration of mitoses of *Pafah1b1*-deficient mouse embryonic fibroblasts (MEFs), using time-lapse live-cell imaging. Through imaging of *Pafah1b1*-deficient MEFs, we found severe defects in cleavage plane positioning and dysregulation of cell membrane contractility. Here we demonstrate that LIS1 determines cleavage plane positioning through a RhoA-actomyosin signaling pathway, defining a novel molecular mechanism of LIS1 underlying spatiotemporal regulation of cell membrane contractility during mitotic cell division.

## Results

### Mislocalization of RhoA and Anillin in *Pafah1b1* mutant neocortical neural progenitor cells (NPCs)

To elucidate molecular mechanisms underlying LIS1-dependent NPC regulation during neocortical development, mitotic phenotypes of *Pafah1b1*-deficient NPCs were analyzed by conducting genetic and immunohistochemical approaches on mouse neocortices derived at embryonic day 14.5 (E14.5) (*Figure 1A–E*). We took advantage of mosaic analysis with double markers on chromosome 11 mouse line (MADM-11; *Hippenmeyer et al., 2010*), since the *Pafah1b1* is located on chromosome 11 away from the centromere. To deplete *Pafah1b1* sparsely in neocortical NPCs during early embryonic development, we first generated *MADM-11^{TG/TG,Pafah1b1}* (TG: tdTomato-GFP fusion) mice co-expressing the heterozygous *Pafah1b1* knock-out (KO) allele. These mice were mated with *MADM-11^{GT/GT}*;*Emx1^{Cre/+}* (GT: GFP-tdTomato fusion) to generate the experimental mosaic animals which carry sparsely labeled NPCs with different expression levels of LIS1 (*MADM-11^{GT/TG,Pafah1b1}*;*Emx1-Cre/+*, abbreviated as *Pafah1b1-MADM*) (*Figure 1B*). In this line, we identified three distinct subpopulations of labeled NPCs with different LIS doses; *Pafah1b1^{+/+}* (red, labeled with tdTomato, 100%

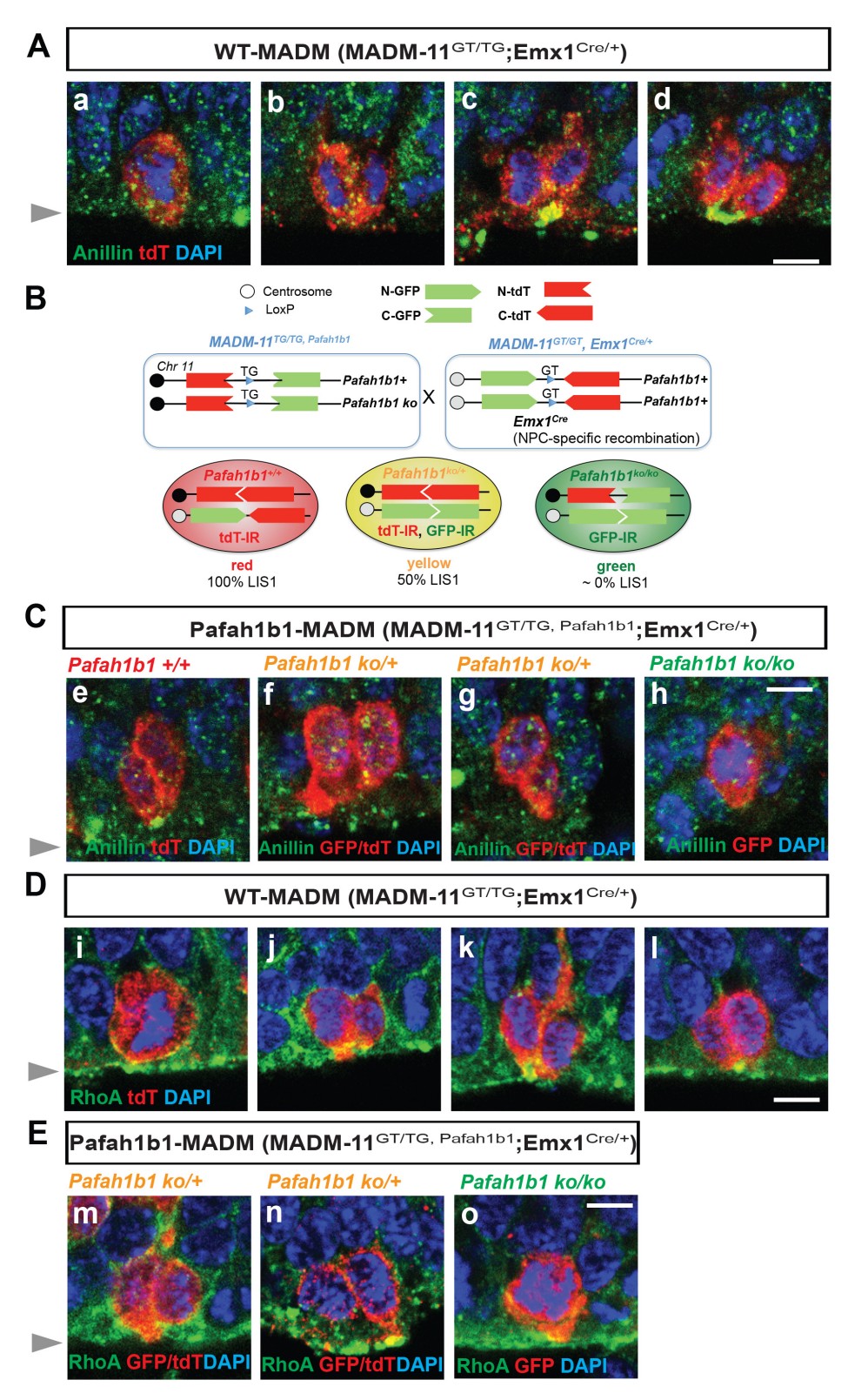

**Figure 1.** Cleavage plane formation and positioning in the neocortical neural progenitor cells (NPCs) in WT-*MADM* and *Pafah1b1-MADM* embryos. (A) Wild-type (WT) NPCs displayed recruitment of Anillin to the basal equatorial cortex and ultimately the Anillin-ring moved to the apical surface of the ventricular zone, forming a 'U'-like shape. (B) Schematic representation of *Pafah1b1-MADM* mating scheme and three types of neocortical NPCs with different LIS1 expression levels. Immunoreactivity (IR) from immunohistochemistry experiment with anti-GFP and anti-tdT-c-Myc antibodies was

*Figure 1 continued on next page*

*Figure 1 continued*

indicated. (C) (e) Midbody-associated Anillin localization in WT (*Pafah1b1*$^{+/+}$) NPCs, (f,g) Midbody-associated Anillin distribution was not detected in *Pafah1b1* heterozygous (*Pafah1b1*$^{ko/+}$) NPCs. (h) Complete knock-out (KO) of *Pafah1b1* (*Pafah1b1*$^{ko/ko}$) in NPCs results in mitotic arrest at the prometaphase- or metaphase-like time-points. (D) (i,j). WT NPCs displayed apical membrane-associated RhoA during cytokinesis. (k,l). RhoA was also co-localized with the midbody, at the future cleavage furrow. (E) (m,n). The cytosol from only one daughter cell retained RhoA-positive puncta in WT NPCs. (o) Complete KO of *Pafah1b1* (*Pafah1b1*$^{ko/ko}$) resulted in detachment of NPCs away from the RhoA-positive apical membrane. Gray arrowheads: ventricular surface (apical). Scale bars: 5 μm. Quantitative data were included in *Figure 1—source data 1*.

The online version of this article includes the following source data for figure 1:

**Source data 1.** Quantification of apical NPCs (RGs).

LIS1 wild-type (WT) levels), *Pafah1b1*$^{ko/+}$ (yellow, double positive for GFP and tdTomato, 50% LIS1 WT levels), and *Pafah1b1*$^{ko/ko}$ (green, labeled with GFP, 0% LIS1) NPCs in an unlabeled *Pafah1b1*$^{ko/+}$ heterozygous background. The fluorescence of each cell enabled us to distinguish the genotype of each cell. The same mating scheme was used to generate WT control animals (*MADM-11*$^{GT/TG}$; *Emx1*$^{Cre/+}$, abbreviated as WT-*MADM*) where green, red, and yellow cells are all WT expressing normal levels of 100% LIS1.

To determine whether *Pafah1b1* deficiency in neocortical NPCs results in displacement of the mitotic cleavage plane with abnormal distribution of contractile components, we assessed Anillin distribution in *Pafah1b1-MADM* neocortices compared with those of WT-*MADM* at E14.5. In the WT-*MADM* neocortex, Anillin was accumulated at the midzone during metaphase-to-anaphase (*Figure 1A–a,b*) and was enriched by forming a 'U' shape (basal-to-apical ingression) at the midbody of NPCs, consistent with previous observations of normal NPC cleavage in WT mice (*Kosodo et al., 2008*; *Figure 1A–c,d*). In the *Pafah1b1-MADM* neocortices, the tdTomato-positive WT NPCs (red, *Pafah1b1*$^{+/+}$) similarly displayed a normal distribution of Anillin at the apical cleavage furrow between the two daughter cells (*Figure 1C–e*). The tdTomato- and GFP-double-labeled *Pafah1b1* heterozygous NPCs (yellow, *Pafah1b1*$^{ko/+}$) did not display an Anillin-rich ring associated with the midbody (*Figure 1C–f,g*). The *Pafah1b1-MADM* (*Figure 1B–C*) neocortex displayed a profound decrease in GFP-positive *Pafah1b1* homozygous KO apical NPCs located at the ventricular zone (green, *Pafah1b1*$^{ko/ko}$). These apical *Pafah1b1*$^{ko/ko}$ NPCs were mostly found at prometaphase or metaphase and located at the ventricular surface with no obvious cell membrane-associated Anillin with dispersed patterns (*Figure 1C–h*), probably due to mitotic arrest after complete loss of LIS1 (*Yingling et al., 2008*). Abnormal distribution of Anillin in *Pafah1b1* mutant NPCs (*Pafah1b1*$^{ko/+}$) implies that LIS1 is an important cell determinant for neocortical NPC cell cleavage and daughter cell separation during cytokinesis in a cell-intrinsic manner.

In apical NPCs of the neocortex, the highest expression of RhoA was detectable at the ventricular surface where NPC cleavage furrows were found (*Gauthier-Fisher et al., 2009*; *Lian et al., 2019*). Overexpression of dominant-negative forms of RhoA in mouse NPCs in vitro induced mislocalization of cleavage furrow with diffuse and dispersed contractile ring (*Lian et al., 2019*), suggesting that RhoA dictates cytokinetic progression and cleavage furrow specification. In the present study, cell membrane-bound forms of RhoA in apical NPCs were identified by fixing the embryonic brains with a 10% trichloroacetic acid (TCA) solution. In the control WT-*MADM* neocortex (*MADM-11*$^{GT/TG}$; *Emx1*$^{Cre/+}$), tdTomato illuminated the cytoplasm of dividing NPCs. At metaphase, RhoA was associated with the apical cell membrane of WT NPCs proximal to the ventricular surface (*Figure 1D–i*). At telophase, RhoA was enriched at the midbody located at the ventricular surface, indicating separation of the two daughter cells (*Figure 1D* j,k,l). However, in the *Pafah1b1-MADM* neocortex (*MADM-11*$^{GT/TG,Lis1}$;*Emx1*$^{Cre/+}$), tdTomato- and GFP-double-labeled *Pafah1b1* heterozygous NPCs (yellow, *Pafah1b1*$^{ko/+}$) displayed an unequal distribution of RhoA skewed to the cytoplasm of only one daughter cell (*Figure 1E–m,n*). The GFP-positive *Pafah1b1* KO NPCs (green, *Pafah1b1*$^{ko/ko}$) were sparsely found at the prometaphase or metaphase with apical membrane-associated RhoA (*Figure 1E–o*). Together, mislocalization of Anillin and RhoA in *Pafah1b1* mutant neocortical NPCs (*Pafah1b1*$^{ko/+}$) indicates that LIS1 may contribute to cleavage plane specification and positioning during late stages of apical NPC mitosis.

## Mislocalization of Anillin in apical NPCs from *Pafah1b1* heterozygous neocortex

We next asked whether *Pafah1b1* heterozygosity leads to changes in cytoarchitecture of the apical NPC niche at the ventricular surface of the neocortex. We deleted one copy of *Pafah1b1* in neocortical NPCs by mating *Pafah1b1* conditional knock-out (CKO) line with the *GFAP-Cre* line (*Zhuo et al., 2001*). In control neocortex (*Pafah1b1*<sup>hc/+</sup> without Cre, hc: hypomorphic conditional), NPCs undergoing vertical divisions (with a vertical cleavage plane angle) possessed basally located Anillin that gradually ingressed apically with the formation of the midbody, defining final location of cleavage furrow (basal-to-apical ingression) (*Figure 2A*), consistent with previous findings (*Kosodo et al., 2008*). NPCs undergoing horizontal or oblique divisions maintained a straight line of Anillin-rich immunoreactive signal in the junctional plate between two daughter cells (*Figure 2B*), suggesting normal cleavage furrow formation at the midzone between the two daughter cells. There were reduced numbers of NPCs undergoing vertical divisions (42%) in NPC-specific *Pafah1b1* heterozygous mutant neocortex (*GFAP-Cre; Pafah1b1*<sup>hc /+</sup>) compared with those numbers (84%) in control neocortex (*Pafah1b1*<sup>hc/+</sup>), consistent with our previous findings (*Yingling et al., 2008*). In addition, NPCs undergoing oblique and horizontal divisions in *Pafah1b1* heterozygous mutant neocortex displayed a diffuse pattern of Anillin associated with similar enrichment at both the apical and basal cell membrane. Only one apically attached daughter cell retained concentrated Anillin-positive puncta at the ventricular surface (*Figure 2B–f,g,i*), suggesting that *Pafah1b1* heterozygosity resulted in Anillin mislocalization during late mitosis of apical NPCs.

## An imbalance in symmetric versus asymmetric NPC divisions after dose-dependent reduction of LIS1

We hypothesized that *Pafah1b1* mutant NPCs may display an imbalance in symmetric vs. asymmetric divisions due to defects in cleavage furrow patterning and cell membrane contractility. We previously reported that neocortical NPCs in *Pafah1b1* mutants had reduced mitotic spindle angles because more cells displayed abnormal cleavage planes that were skewed to oblique and horizontal divisions (*Yingling et al., 2008*). Here, we analyzed neocortical NPC divisions by tracing the inheritance of atypical protein kinase C zeta (aPKC, PKCζ), an apical complex component, into each daughter cell. The cleavage furrow was identified by the location of Cadherin hole in the continuous line of N-Cadherin that indirectly indicates the contractile ring positioning (*Marthiens and ffrench-Constant, 2009*). The relative positioning of the mitotic spindle was defined by the midline between two chromosome sets. Thus, we determined that neocortical NPCs from *Pafah1b1*-deficient mutants (*Pafah1b1*<sup>hc/ko</sup>) more frequently displayed unequal inheritance of aPKCζ to the daughter cells (*Figure 3A–C*). In addition, apical NPCs in *Pafah1b1*<sup>hc/ko</sup> neocortices possessing horizontal cleavage planes displayed less polarized aPKCζ along the apical-basal axis (*Figure 3B*), indicating mild alterations of apical polarity in *Pafah1b1*-deficient neocortical NPCs. In *Pafah1b1*<sup>hc/+</sup> control neocortices, 30.7% of NPCs displayed unequal distribution of aPKCζ into the cytoplasm of the two daughter cells (presumably asymmetric divisions). However, in *Pafah1b1*<sup>hc/ko</sup> brains, 68.8% of NPCs displayed unequal distribution (*Figure 3C*). These results suggest that *Pafah1b1* deficiency in neocortical NPCs results in abnormal cleavage furrow positioning that may induce unequal inheritance and distribution of cell fate determinants into two daughter cells that ultimately leads to an increase in asymmetric (presumably neurogenic) NPC divisions.

## Perturbed daughter cell separation and an increase in binucleation during mitosis of *Pafah1b1* mutant MEF

We also tested whether *Pafah1b1* deficiency impairs cleavage plane positioning and completion of mitosis in MEFs using time-lapse microscopy. We visualized mitotic events of late stages of mitosis of MEFs in vitro in greater detail than we could do in the neocortex in vivo. We examined the frequency of normal daughter cell separation with complete abcission from both WT MEFs (*Pafah1b1*<sup>+/+</sup>) and mutant *Pafah1b1* compound heterozygous MEFs (*Pafah1b1*<sup>hc/ko</sup>) expressing 35% of LIS1 compared to normal WT levels (*Yingling et al., 2008*). Sequential mitotic events were monitored by time-lapse live-cell imaging using mCherry-α-Tubulin, a fluorescently labeled MT marker, and histone 2B (H2B)-GFP, a chromosomal probe (*Moon et al., 2014*). A majority of WT MEFs displayed successful daughter cell separation (85.4 ± 6.1%), while in *Pafah1b1*<sup>hc/ko</sup> MEFs, this frequency was decreased

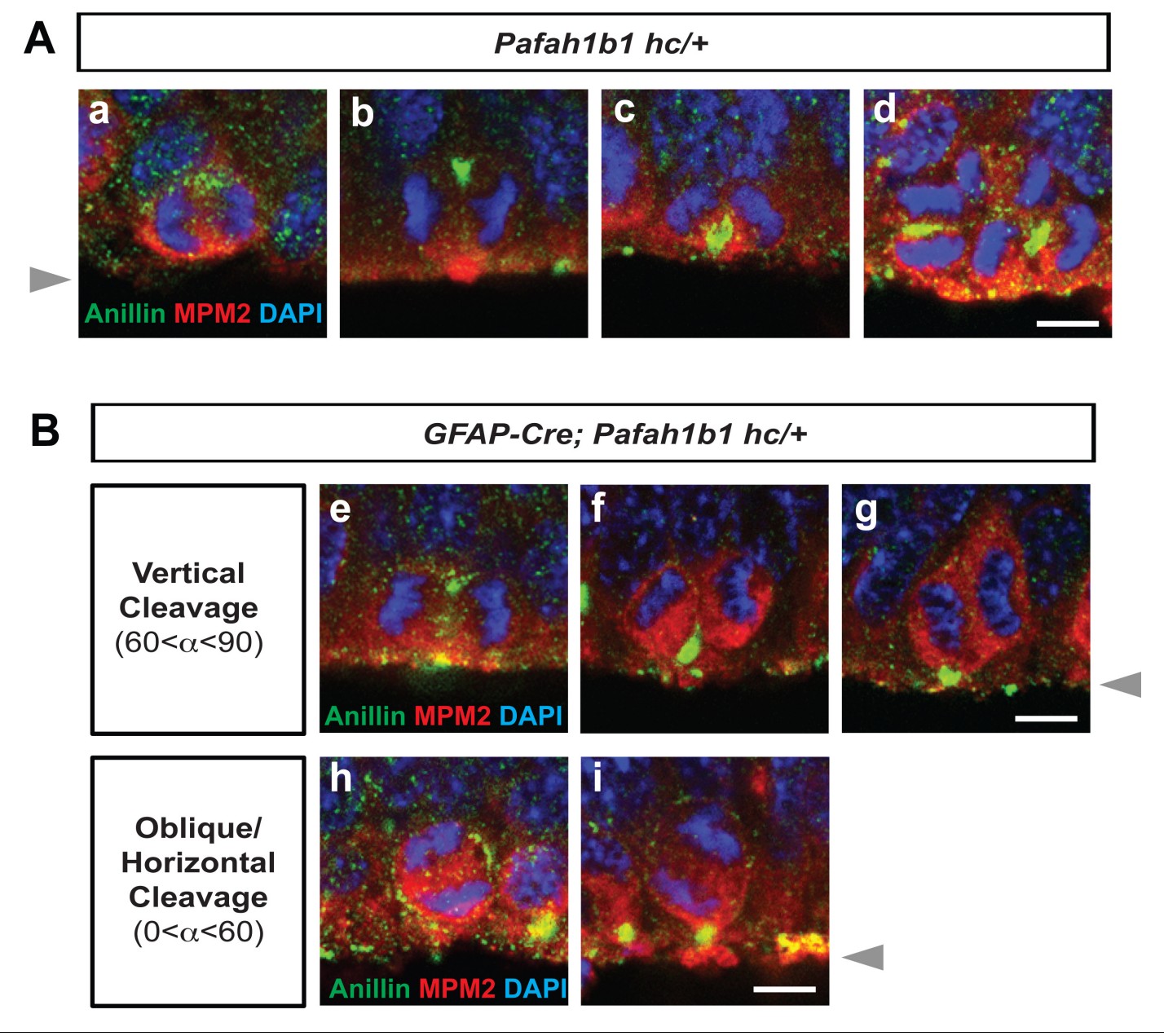

**Figure 2.** Localization of Anillin-ring in neocortical NPCs from controls and *GFAP-Cre*-induced *Pafah1b1* mutant mouse embryos. (**A**) Normal Anillin-ring distribution during cytokinesis of *Pafah1b1* control (*Pafah1b1hc/+*) NPCs. hc: hypomorphic conditional allele, (a) Early anaphase, (b) Mid-anaphase with a basally located Anillin-ring, (c) 'U'-like shape Anillin-ring at the cleavage furrow, (d) Midzone-specific Anillin localization. (a,b,c) Vertical cleavage plane (60°< α < 90°), α: spindle angle compared to the ventricular surface, (d) Oblique and horizontal cleavage plane (0°< α < 60°). (**B**) Abnormal Anillin-ring localization during cytokinesis of *Pafah1b1*-deficient mutant NPCs (*GFAP-Cre; Pafah1b1hc/+*). (e) Basal and apical Anillin-rings at the equator of NPCs, (f,g) Moderately skewed Anillin-ring localization in only one daughter cell. (h) The equator-associated Anillin-ring was detected with tilted spindle angle in *Pafah1b1*-deficient NPCs. (i) *Pafah1b1*-deficient mutant NPCs displayed horizontal cleavage plane with apical memrbane-associated Anillin puncta. Gray arrowheads: ventricular surface (apical). Scale bars: 5 μm. Quantitative data were included in *Figure 2—source data 1*.

The online version of this article includes the following source data for figure 2:

**Source data 1.** Quantification of apical NPCs (RGs).

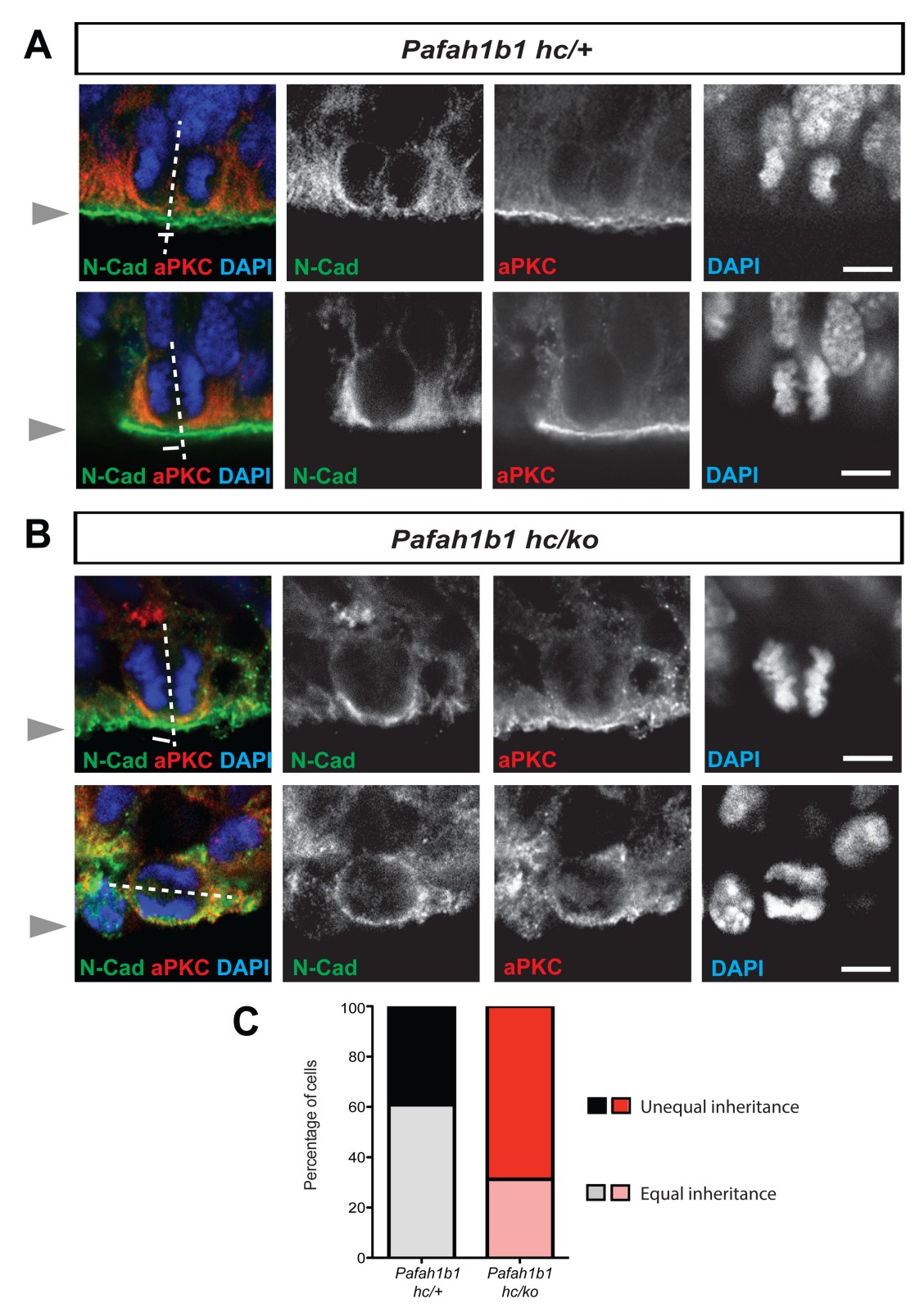

**Figure 3.** Alterations in symmetric and asymmetric divisions determined by equal vs. unequal inheritance of atypical PKC (aPKC, PKCζ) and Cadherin holes. (**A**) Normal cytokinesis in neocortical NPCs from *Pafah1b1* controls (*Pafah1b1$^{hc/+}$*). Upper panel: Control NPCs displayed vertical cleavage with equal inheritance of aPKC and Cadherin hole. Lower panel: Control NPCs displayed vertical cleavage with unequal inheritance of aPKC and Cadherin hole – presumably reflecting an asymmetric neurogenic division. (**B**) Upper panel: *Pafah1b1*-deficient NPCs (*Pafah1b1$^{hc/ko}$*) displayed vertical cleavage

*Figure 3 continued on next page*

*Figure 3 continued*

with unequal inheritance of aPKC and Cadherin hole. Lower panel: *Pafah1b1*-deficient NPCs displayed horizontal cleavage with unequal and very skewed inheritance of aPKC and Cadherin hole. This *Pafah1b1*-deficient NPC retained less polarized aPKC/N-Cadherin protein distribution (less apical enrichment of aPKC and N-Cadherin) along the apical-basal axis. (C) Disturbances in the frequencies of equal vs. unequal inheritance of cell fate determinants in *Pafah1b1*-deficient NPCs. Gray arrowheads: ventricular surface (apical). Scale bars: 5 µm. Quantitative data were included in *Figure 3— source data 1*.

The online version of this article includes the following source data for figure 3:

**Source data 1.** Quantification of aPKCζ inheritance in apical NPCs (RGs).

(18.7 ± 8%) (*Figure 4A*). By contrast, binucleated daughter cells were increased in *Pafah1b1*$^{hc/ko}$ MEFs (40.1 ± 1.2%) relative to WT MEFs (5.1 ± 3.1%) (*Figure 4B*). To assess whether an increase in incomplete cell separation leads to changes in apoptotic cell death, we immunostained cleaved caspase-3. *Pafah1b1*$^{hc/ko}$ MEFs had an increased number of cleaved caspase-3-immunoreactive cells compared with WT, suggesting that *Pafah1b1* deficiency induces more frequent apoptosis of MEFs (*Figure 4C*). We also confirmed an increase in cell death (from 3- to 6-fold) from live-cell imaging of *Pafah1b1*$^{hc/ko}$ MEFs.

To determine whether acute loss of *Pafah1b1* also induces defective cell separation, we derived MEFs containing CKO alleles of *Pafah1b1* and tamoxifen (TM)-inducible *CAGG-CreERT2* line (*Hayashi and McMahon, 2002*) with the genotype *CAGG-CreERT2; Pafah1b1*$^{hc/hc}$. We treated these MEFs with 4-hydroxy-TM for 12 hr and compared mitotic phenotypes with control MEFs (*CAGG-CreERT2; Pafah1b1*$^{+/+}$). Similar to *Pafah1b1*$^{hc/ko}$ mutant MEFs, acute deletion of *Pafah1b1* in TM-treated *Pafah1b1* CKO MEFs (*CAGG-CreERT2; Pafah1b1*$^{hc/hc}$ +TM$^{12\ h}$) induced a decrease in normal daughter cell separation (36.6 ± 5.3%) compared with control levels (69.4 ± 4.3%) (*Figure 4D*). Consistently, a trend of frequent daughter cell separation failure and higher percentage of binucleation (25.8 ± 6.4%) than controls (*CAGG-CreERT2; Pafah1b1*$^{+/+}$ +TM$^{12\ h}$) (15.7 ± 6.1%) was observed from live-cell imaging (*Figure 4E*) but this was not statistically significant.

## Cell shape oscillation and mitotic spindle rocking during mitosis of *Pafah1b1* mutant MEFs

We analyzed movements of MT-enriched midbody and cleavage plane positioning throughout the entire mitotic cell cycle by live-cell imaging. Acute deletion of *Pafah1b1* in MEFs (*CAGG-CreERT2; Pafah1b1*$^{hc/hc}$ +TM$^{12\ h}$) resulted in drastic cell shape oscillation (*Figure 5A*, *Video 1*), similar to the phenotypes described in Anillin-depleted cells (*Echard et al., 2004*; *Kechad et al., 2012*; *Piekny and Glotzer, 2008*; *Straight et al., 2003*; *Zhao and Fang, 2005*). At the anaphase onset of *Pafah1b1* mutant MEFs, the midbody was properly positioned in the central spindle zone. However, the position of the midbody was gradually destabilized and began vigorously oscillating between the two daughter cells. Surprisingly, the initially separated chromosome sets moved back and forth between two daughter cells and mitotic spindle rocking was prominent throughout this process. This phenotype was previously termed 'hyper-contractility' in cells with depleted contractile ring components, and is caused by hyper-activity of the actomyosin during cytokinesis (*Werner and Glotzer, 2008*). Surprisingly, in one example seen in *Pafah1b1* mutant MEFs (*Figure 5B*, *Video 2*), one daughter cell inherited two-sets of chromosomes with binucleation while the other daughter cell underwent cell death processes.

We next examined the incidence of the hyper-contractility phenotypes during MEF mitosis. WT MEFs rarely displayed oscillated cell membrane movements (3.6 ± 3.6%). By contrast, more than half of the mitosis (56.2 ± 8.4%) in *Pafah1b1*$^{hc/ko}$ MEFs exhibited the hyper-contractility phenotypes (*Figure 5C*). The incidence of ectopic membrane bulges was increased in *Pafah1b1* mutant MEFs which prompted us to monitor cell shape changes and cell membrane dynamics by merging phase-contrast images in live-cell imaging analysis. We found that *Pafah1b1*$^{hc/ko}$ MEFs frequently showed cell membrane blebbing (*Figure 5D*) and the size of ectopic membrane blebs was much larger than those in WT MEFs. In accordance with these findings, the hyper-contractility phenotypes with cell membrane blebbing were more frequent in *Pafah1b1* CKO MEFs (*CAGG-CreERT2; Pafah1b1*$^{hc/hc}$ +TM$^{12\ h}$) (40.5 ± 8.4%) compared with control MEFs (*CAGG-CreERT2; Pafah1b1*$^{+/+}$ +TM$^{12\ h}$) (13.5 ± 2.4%) (*Figure 5E*).

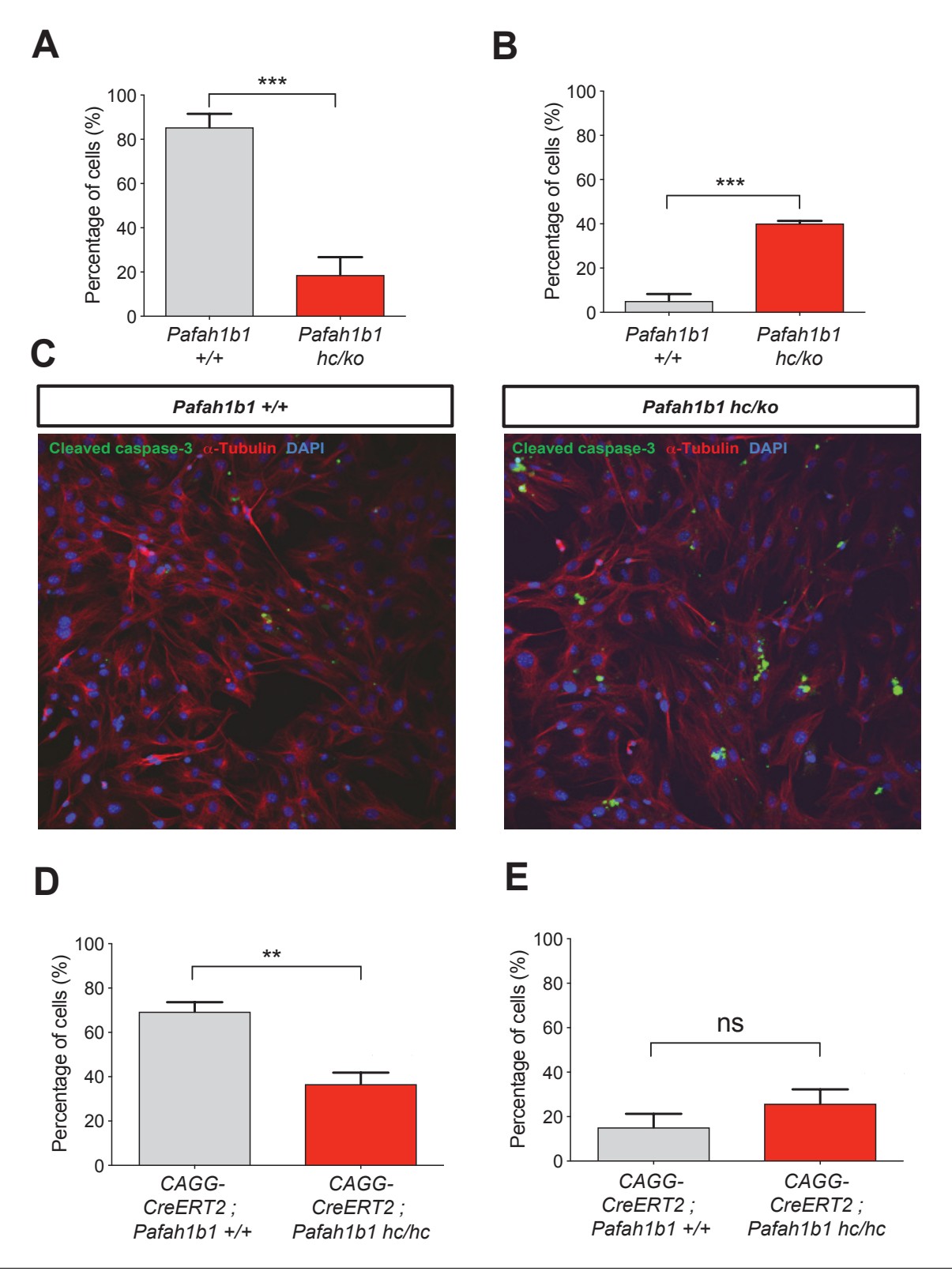

**Figure 4.** Loss of *Pafah1b1* in mouse embryonic fibroblasts (MEFs) impairs normal cytokinesis and leads to binucleation of the daughter cells. (**A**) *Pafah1b1* mutant mouse embryonic fibroblasts (MEFs) (*Pafah1b1*<sup>hc/ko</sup>) displayed failure of cytokinesis, daughter cell separation, compared with WT MEFs, analyzed by time-lapse live-cell imaging. (**B**) Increased binucleation by fused daughter cells from MEFs with acutely deleted *Pafah1b1* (*CAGG-CreERT2; Pafah1b1*<sup>hc/hc</sup>). (**C**) Increased apoptotic cell death from *Pafah1b1*<sup>hc/ko</sup> MEFs compared with WT MEFs (green: cleaved caspase-3, red: α-

*Figure 4 continued on next page*

*Figure 4 continued*

Tubulin, blue: DAPI). (**D**) Acutely induced *Pafah1b1* conditional knock-out (CKO) mutant MEFs (*CAGG-CreERT2; Pafah1b1hc/hc* + TM12h) also displayed less frequency of normal daughter cell separation compared with control MEFs (*CAGG-CreERT2; Pafah1b1+/+* + TM12h). (**E**) An increased trend in binucleation was found in *Pafah1b1* CKO mutant MEFs relative to control MEFs. However, it did not reach significant. Scale bars: 50 µm. Quantitative data were included in *Figure 4—source data 1*.

The online version of this article includes the following source data for figure 4:

**Source data 1.** Quantification of MEFs.

## Mislocalization of RhoA and contractile ring components in mitosis of *Pafah1b1* mutant MEFs

We tested whether critical regulators of cytokinesis contribute to *Pafah1b1*-deficiency-induced cell membrane hyper-contractility. Since RhoA is an important master GTPase regulating cytokinesis, we examined RhoA localization in *Pafah1b1* mutant MEFs fixed with 10% TCA (*Yonemura et al., 2004*). In WT MEFs, the active forms of RhoA symmetrically accumulated at both sides of the equatorial cortex and were concentrated in the midbody at late telophase and during cytokinesis (*Figure 6A*). From anaphase to telophase, RhoA colocalized with the cell membrane-associated Anillin, a key component of the contractile ring that defines the cleavage furrow ingression site. By contrast, 60% of *Pafah1b1hc/ko* MEFs displayed mislocalization of the cleavage furrow with the formation of large polar blebs in ectopic membrane sites (*Figure 6B*). An aberrant cleavage furrow was found at only one side of the cell equatorial cortex of *Pafah1b1hc/ko* MEFs, resulting in asymmetric mislocalization of RhoA and Anillin (*Figure 6B*, upper panel). The binucleated *Pafah1b1hc/ko* MEFs displayed protruding polar blebs in ectopic locations (*Figure 6B*, lower panel). Intriguingly, RhoA immunoreactive signal decorated the cell membrane of all of these polar blebs, suggesting that RhoA and actomyosin may be the primary signals to produce aberrant membrane bulges in *Pafah1b1hc/ko* MEFs.

Since the LIS1-dynein-dynactin complex is associated with the cortical cell membrane (*Faulkner et al., 2000*; *Moon et al., 2014*), it is plausible that cortical dynactin, P150glued, may be less enriched in the cell membrane and cause Anillin mislocalization in *Pafah1b1hc/ko* MEFs. Cortical P150glued was incorporated within the polar cortex but it was excluded in the cleavage furrow where Anillin was associated with the equatorial cortex in WT MEFs (*Figure 6C*). However, reciprocal exclusion of polar P150glued vs. equatorial Anillin was less obvious in *Pafah1b1hc/ko* MEFs (*Figure 6D*; *Moon et al., 2014*). From anaphase to telophase, the cells with cleavage furrow-associated Anillin were reduced in *Pafah1b1hc/ko* MEFs (27.1%) relative to WT control *Pafah1b1+/+* MEFs (72.3%) (*Figure 6E*, in gray).

We found that *Pafah1b1*-deficient MEFs had mislocalization of a key contractile ring component, Anillin. Hence, we examined cellular localization of LIS1 and Anillin during late stages of mitosis and cytokinesis and evaluated co-localization between LIS1 and Anillin. However, only rabbit-derived antibodies against LIS and Anillin were sufficiently specific in MEFs, so we were not able to perform direct dual immunocytochemistry with combination of LIS1 and Anillin antibodies. When we alternatively traced LIS1 and Anillin antibody separately with α-Tubulin, we found no evidence of substantial overlap of intercellular compartments with LIS1-Anillin in MEFs undergoing late mitosis and cytokinesis. During mitotic division of WT MEFs, LIS1 was localized only to the centrosomes in metaphase-to-anaphase and the central MTs during cytokinesis displaying mostly cytosolic distribution (*Figure 7A*), while Anillin was initially seen on the lateral cortex and later specifically accumulated in the cytokinetic abcission ring between two central MT bundles, near the equatorial cortex (*Figure 7B*). These results support that LIS1 may only indirectly regulate Anillin through F-actin cytoskeleton regulation.

## Mislocalization of F-actin and myosin II in mitosis of *Pafah1b1* mutant MEFs

A previous study demonstrated that the leading process-associated F-actin is misregulated during migration of post-mitotic neurons in *Pafah1b1ko/+* mutants (*Kholmanskikh et al., 2003*). Actin dysfunction induced by loss of LIS1 may lead to cytokinetic defects and hyper-contractility seen in *Pafah1b1hc/ko* MEFs. To test this possibility, we co-immunostained MEFs with phalloidin (F-actin zone) and dynein intermediate chain (DIC 74.1). WT MEFs had an F-actin zone located at the

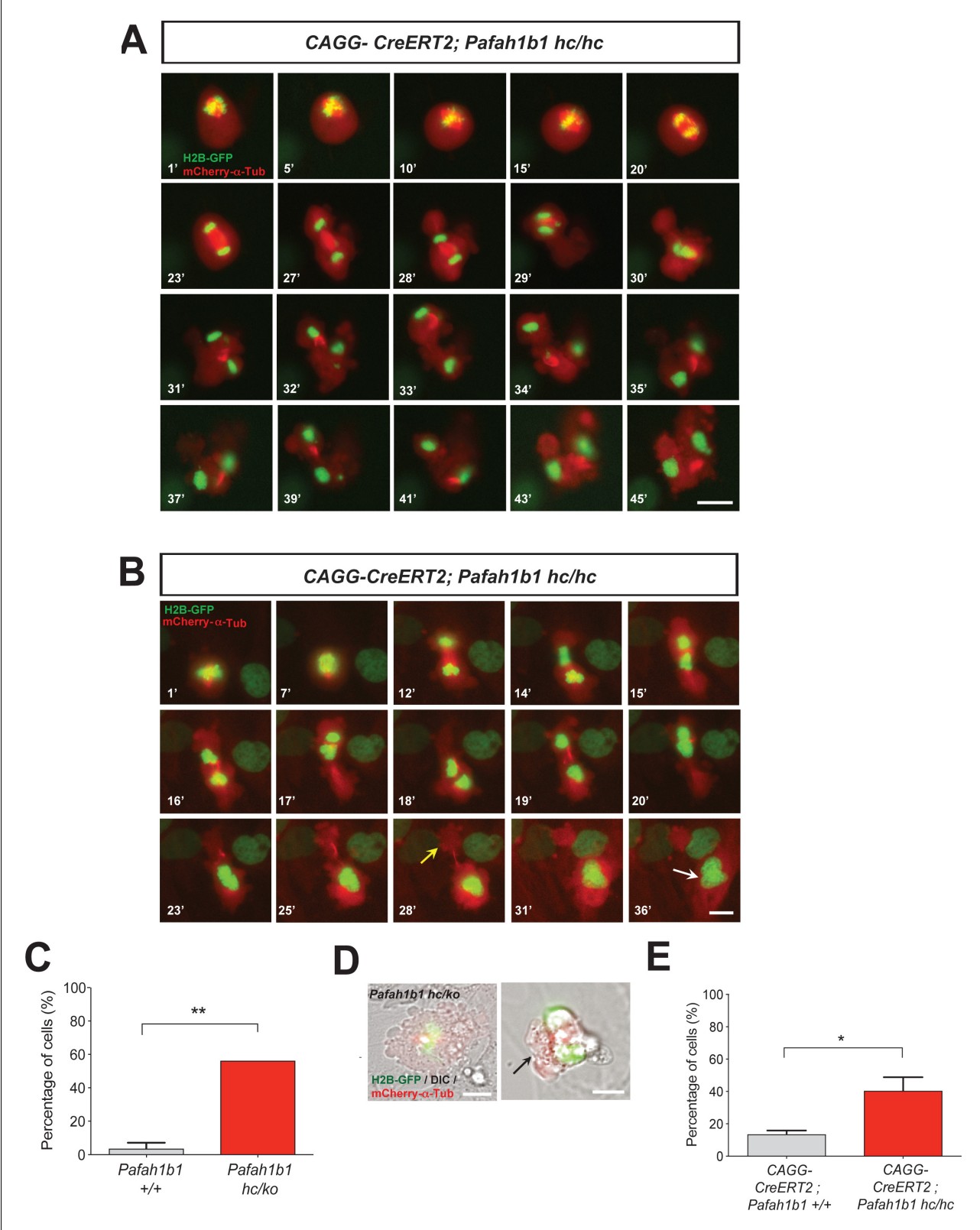

**Figure 5.** Loss of *Pafah1b1* results in vigorous cell shape oscillation and mitotic spindle rocking that ultimately leads to cytokinetic failure with aberrant cleavage plane positioning. (**A**) Acutely deleted *Pafah1b1* mutant MEFs (*CAGG-CreERT2; Pafah1b1hc/hc* + TM12h) displayed severe cleavage plane defects during cytokinesis. There was instability of the midbody (highly concentrated with red fluorescence: mCherry-α-Tubulin) accompanied by cell shape oscillation and spindle rocking (green: H2B-GFP). Scale bar: 20 μm. (**B**) The MEFs with acutely deleted *Pafah1b1* displayed abnormal cell shape

*Figure 5 continued on next page*

*Figure 5 continued*

and chromosome oscillation between two daughter cells. Yellow arrow (28'): anucleated one daughter cell, White arrow (36'): binucleated and bi-lobed daughter cell. Scale bar: 10 μm. (C) An increase in the frequency of hyper-contractility phenotype in *Pafah1b1*[hc/ko] MEFs. (D) Single snapshots from the merged images with DIC (differential interference contrast) and fluorescence images (green: H2B-GFP, red: mCherry-α-Tubulin, gray: DIC) from time-lapse movies (black arrows: aberrant formation of huge polar blebs). (E) Hyper-contractility quantification indicates that *Pafah1b1* CKO mutant MEFs have increases in cell membrane blebbing phenotypes than control MEFs. Scale bars: 10 μm. Quantitative data were included in *Figure 5—source data 1*.

The online version of this article includes the following source data for figure 5:

**Source data 1.** Quantification of MEFs.

---

midbody that also colocalized with DIC (*Figure 7A*, upper panel). By contrast, *Pafah1b1*[hc/ko] MEFs had wider F-actin cortical patches at the midzone that extended to the polar cortex in one daughter cell compared with WT MEFs. Central MT-associated F-actin fibers were also evident in *Pafah1b1*[hc/ko] MEFs (*Figure 7A*, lower panel).

We next examined the localization of Myosin II in *Pafah1b1*[hc/ko] MEFs relative to WT MEFs. Myosin II is the main molecular motor of the cortical contraction complex of cytokinesis, cooperatively working with the F-actin (*Piekny et al., 2005*). In WT MEFs, non-muscle myosin heavy chain II A (NMHCIIA) immunoreactive signal at the spindle midzone overlapped with the F-actin focus zone (*Figure 8B*, upper panel). By contrast, we found that MHCIIA distribution was diffused and dispersed at the spindle midzone in *Pafah1b1*[hc/ko] MEFs (*Figure 8B*, lower panel). Quantitation of dispersed MHCIIA and F-actin distribution indicated that *Pafah1b1*[hc/ko] MEFs had fewer cells with normal cleavage furrow-associated actomyosin patterns (*Pafah1b1*[+/+], 73.4% vs. *Pafah1b1*[hc/ko], 29.7% - in gray). Conversely, *Pafah1b1*[hc/ko] MEFs showed an increase in cells with ectopic distribution of actomyosin (*Pafah1b1*[+/+], 26.6% vs. *Pafah1b1*[hc/ko], 70.3% - in red) (*Figure 8C*). These findings suggest that cortical constriction at the equatorial cortex was impaired in *Pafah1b1*[hc/ko] MEFs, leading to mispositioning of the cleavage furrow by actomyosin dysfunction.

## Abnormal Myosin II movements during mitosis of *Pafah1b1* mutant MEFs

We also monitored actomyosin dynamics by performing time-lapse live-cell imaging of non-muscle myosin regulatory light chain1 (MRLC1)-GFP in MEFs. MRLC1-GFP is a reliable molecular probe to illustrate the localization of the Myosin II (*Beach et al., 2011*; *Miyauchi et al., 2006*). We infected MEFs with MRLC1-GFP and H2B-tdTomato retroviruses to visualize Myosin II and chromosomes, respectively. In control MEFs (*CAGG-CreERT2*; *Pafah1b1*[+/+] +TM[24 h]), MRLC1 accumulated at the equatorial cortex and the contractile ring, then it constricted normally (*Figure 9A*, *Video 3*). By contrast, Cre-inducible *Pafah1b1* CKO mutant MEFs (*CAGG-CreERT2*; *Pafah1b1*[hc/hc] +TM[24 h]) displayed impairments in Myosin II movements (*Figure 9B*, *Video 4*). Cleavage furrow ingression first occurred but regressed abnormally, with failure to restrict the cleavage furrow at the equatorial cortex. Cleavage furrow mispositioning occurred and resulted in chromosome missegregation. Multiple ectopic membrane bulges were found near the polar cortex in *Pafah1b1* mutant MEFs, suggesting that aberrant cytoplasmic pushing forces may be generated by ectopically located Myosin II.

In the severely affected cells, *Pafah1b1* mutant MEFs displayed an uncoupling between

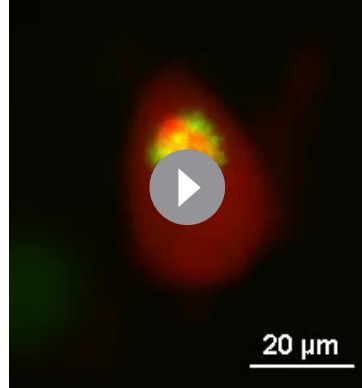

**Video 1.** Cytokinesis defects in *Pafah1b1* mutant MEFs. Time-lapse live cell imaging of mitotic cell division from *Pafah1b1* mutant MEFs (*CAGG-CreERT2*; *Pafah1b1*[hc/hc] +TM[12 h]) treated with 4-hydroxy tamoxifen (TM). H2B-GFP and mCherry-α-Tubulin labeled fluorescence signals were acquired with a 1 min interval by Nikon Ti epifluorescence microscope. During cytokinesis of *Pafah1b1*-deficient MEFs, severe cytokinesis defects were observed such as vigorous cell shape oscillation and spindle rocking.

https://elifesciences.org/articles/51512#video1

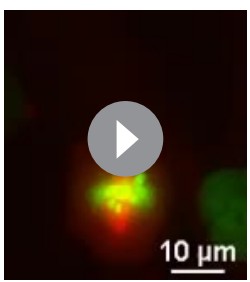

10 µm

**Video 2.** Formation of binucleated daughter cells by cytokinesis failure in *Pafah1b1* mutant MEFs. Time-lapse live cell imaging of mitotic cell division from *Lis1* mutant MEFs (*CAGG-CreERT2; Pafah1b1hc/hc* +TM$^{12 h}$) treated with 4-hydroxy TM. H2B-GFP and mCherry-α-Tubulin labeled fluorescence signals were acquired with a 30 s interval by Nikon Ti epifluorescence microscope. The *Pafah1b1*-deficient MEFs underwent abnormal cytokinesis and resulted in formation of binucleated daughter cells.
https://elifesciences.org/articles/51512#video2

chromosome segregation and cytokinesis (*Figure 9C*, *Video 5*). In normal cytokinesis, chromosome sets first separated into the two daughters, followed by equatorial constriction to initiate cytokinesis. However, this sequential progression of cytokinesis was disrupted in *Pafah1b1* mutant MEFs. Although chromosome sets did not segregate precisely into two daughter cells before the anaphase onset of mitotic spindle elongation, actomyosin-mediated hyper-contractility of the cell membrane triggered cytokinesis progression, producing binucleated cells.

## Failure to maintain contractile ring components at the equatorial cortex in *Pafah1b1* mutant MEFs

To test whether the actomyosin dysfunction in *Pafah1b1* mutant MEFs exert deleterious effects on the recruitment of contractile components to the equatorial cortex, we performed time-lapse live-cell imaging of Septin6 (SEPT6)-GFP (*Gilden et al., 2012*). Like Anillin, the Septin complex is the key component of the contractile ring during cell cleavage. In control MEFs, SEPT6 was enriched at the equatorial cortex and stayed at the cleavage furrow ingression site (*Figure 10A*, *Video 6*). However, after acute *Pafah1b1* deletion in *Pafah1b1* CKO mutant MEFs (*CAGG-CreERT2; Pafah1b1hc/hc* +TM$^{24 h}$), SEPT6 was initially observed at the equatorial cortex but then it regressed, and weak SEPT6 accumulation was detected at the cell equator accompanied by cortical deformation and chromosome oscillation/rocking (chromosomes were identified as darker spots in the SEPT6-GFP background) (*Figure 10B*, *Video 7*). Further live-cell imaging of SEPT6-GFP infected MEFs showed that the frequency of cytokinetic failure with concomitant SEPT6 mislocalization was increased by 4.7-fold in *Pafah1b1* mutant MEFs (66.7% in *Pafah1b1hc/ko*) compared with WT controls (14.3% in *Pafah1b1+/+*) (*Figure 10C*). Intercellular cytokinetic bridges and binucleation events resulted from incomplete cytokinesis were observed only from *Pafah1b1*-deficient mutant MEF, not like WT MEFs. Together, these results indicate that the contractile ring components were not strictly retained at the equatorial cortex, similar to the defective localization of actomyosin, resulting in hyper-contractility and cytokinesis failure in *Pafah1b1* mutant MEFs.

## Modulation of RhoA hyper-activity in WT and *Pafah1b1* mutant MEFs

If RhoA was hyper-activated in *Pafah1b1* mutant MEFs, the cells expressing a constitutively active form of RhoA (CA-RhoA) might mimic cytokinesis defects found in the *Pafah1b1* mutant MEFs. To test this possibility, WT MEFs were infected with retrovirus encoding human CA-RhoA (CA-RhoA)-GFP fusion protein. As expected, CA-RhoA-GFP-infected WT MEFs (*Pafah1b1+/+* + CA-RhoA-GFP) displayed an increased frequency of cytokinesis failure with mislocalization of Anillin, similar to *Pafah1b1hc/ko* MEFs (*Figure 11A*). WT MEFs infected with GFP control retrovirus (*Pafah1b1+/+* + GFP) displayed normal cytokinesis (*Figure 11B*).

Since *Pafah1b1hc/ko* MEFs displayed the mitotic phenotypes resembling RhoA hyper-activity, we hypothesized that inhibition of RhoA by overexpression of a dominant negative form of RhoA (DN-RhoA) may ameliorate cytokinesis defects in *Pafah1b1hc/ko* MEFs. *Pafah1b1hc/ko* MEFs infected with retrovirus encoding a human DN-RhoA (DN-RhoA)-GFP fusion protein (*Pafah1b1hc/ko* + DN-RhoA-GFP) displayed a decreased frequency of cytokinesis defects compared with control MEFs (*Pafah1b1hc/ko* + GFP) (*Figure 11B*). These results imply that *Pafah1b1hc/ko* MEFs displayed hyper-activation of RhoA and inhibition of RhoA that enabled the rescue of cytokinesis defects seen in *Pafah1b1hc/ko* MEFs.

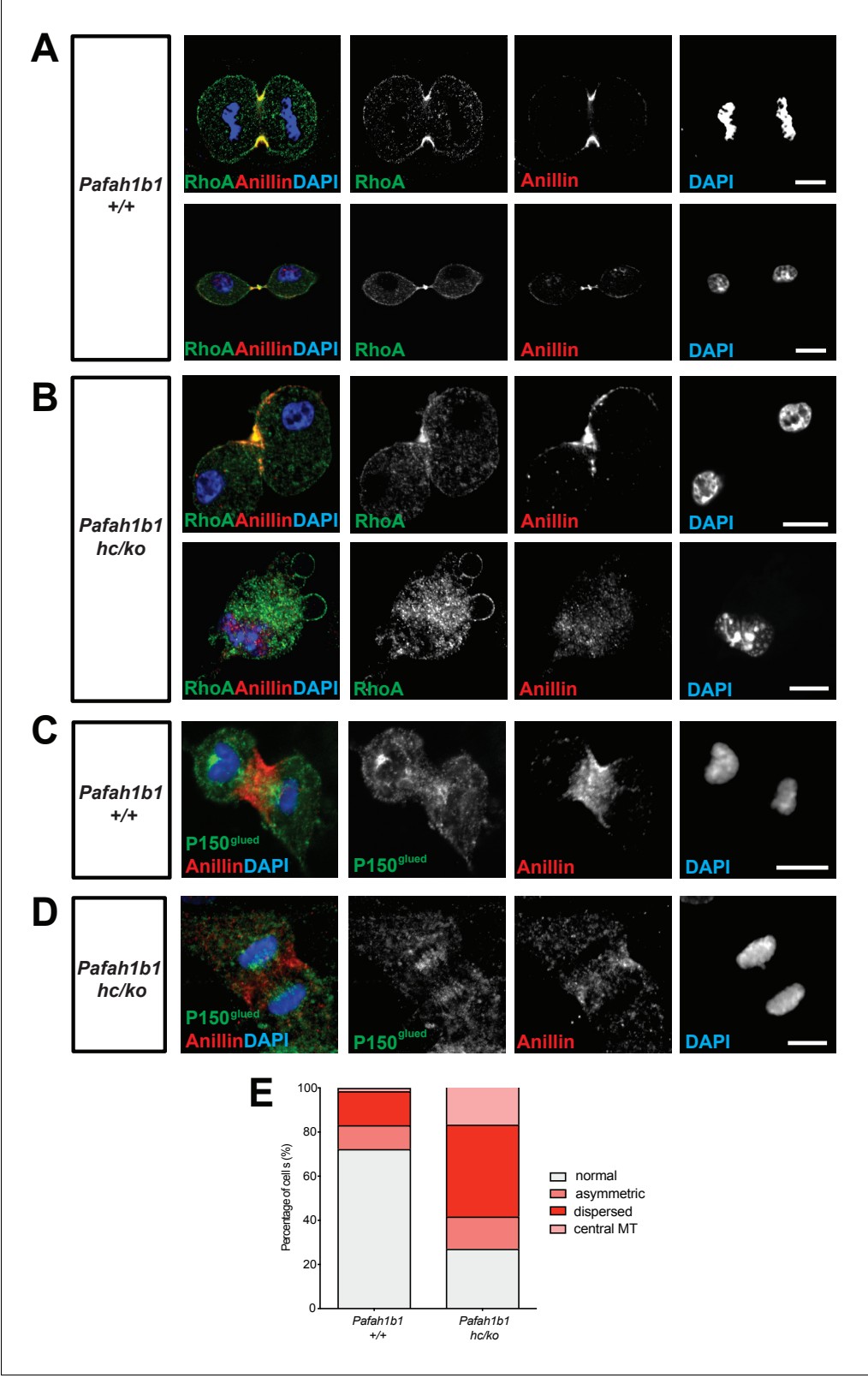

**Figure 6.** *Pafah1b1* mutant MEFs display mislocalization of RhoA and Anillin, an important contractile ring component. (**A**) WT MEFs displayed normal cleavage furrow positioning labeled with TCA-fixed RhoA and Anillin co-staining at the equatorial cortex from early anaphase to telophase. (**B**) *Pafah1b1*^hc/ko^ MEFs displayed mislocalization of RhoA and Anillin. Upper panel: asymmetric cleavage furrow formation. Lower panel: binucleated cells with RhoA-positive aberrant membrane blebs (**A,B**) (green: RhoA, red: Anillin, blue: DAPI). (**C**) WT MEFs displayed an Anillin-positive zone in the cell

*Figure 6 continued on next page*

*Figure 6 continued*

equator. The polar-cortex-associated cortical P150glued dynactin staining pattern was excluded from and did not overlap with the Anillin-ring. (**D**) *Pafah1b1hc/ko* MEFs exhibited reduced cortical P150glued dynactin at the polar cortex and also have decreased and dispersed Anillin distribution to the equatorial cortex. (**C,D**) (green: P150glued dynactin, red: Anillin, blue: DAPI). (**E**) Quantification of normal, asymmetric, dispersed, and central MR-associated Anillin distribution in MEFs. Scale bars: 10 µm. Quantitative data were included in *Figure 6—source data 1*.

The online version of this article includes the following source data for figure 6:

**Source data 1.** Quantification of MEFs.

## Discussion

A number of previous studies have demonstrated that LIS1 is required for neuronal migration and the production of the appropriate numbers of post-mitotic neurons that migrate into their functional cortical layers during embryonic brain development (*Gambello et al., 2003*; *Hirotsune et al., 1998*; *Sasaki et al., 2000*; *Tsai et al., 2005*; *Youn et al., 2009*; *Hippenmeyer et al., 2010*). In addition,

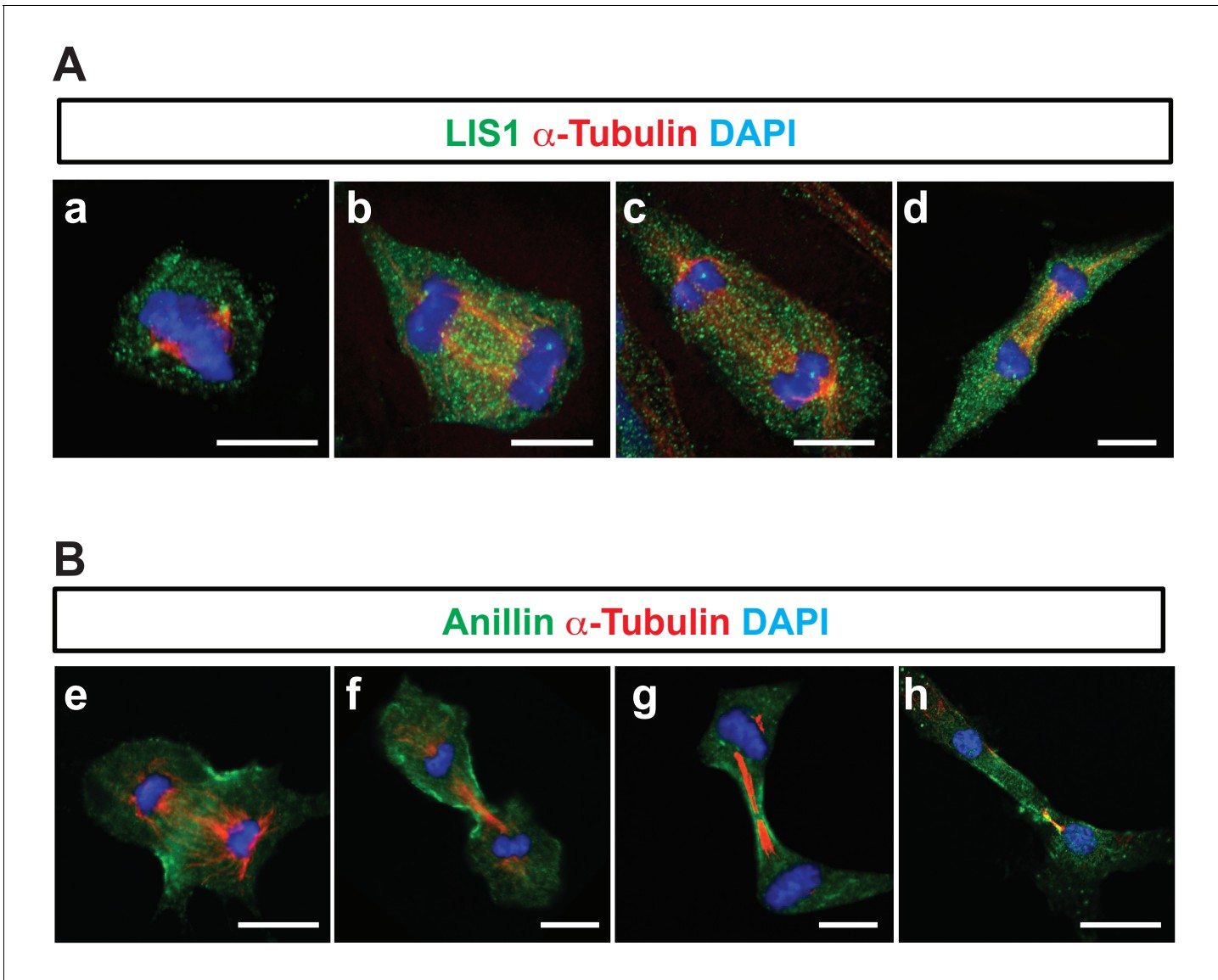

**Figure 7.** Cellular localization of LIS1 and Anillin in MEFs during cytokinesis. (**A**) Wild-type (WT) MEFs were analyzed for immunocytochemistry with Rabbit anti-LIS1 and Rat anti-α-Tubulin. (**B**) With Rabbit anti-Anillin and Rat anti-α-Tubulin. Scale bars = 10 µm.

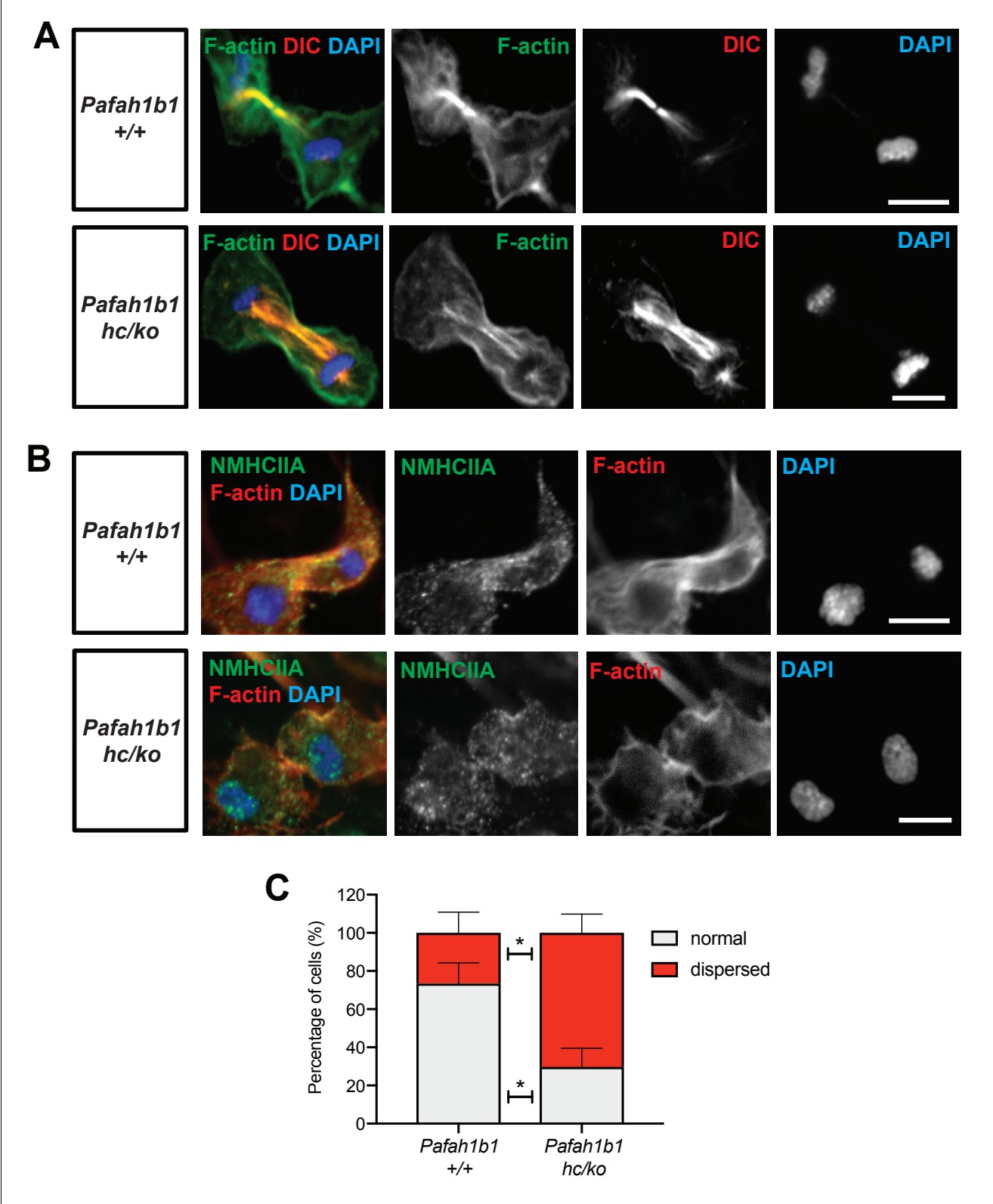

**Figure 8.** *Pafah1b1* mutant MEFs show alterations in F-actin focus zone and Myosin II motor localization during the late stages of mitosis. (**A**) WT MEFs with accumulation of non-muscle myosin heavy chain II-A (NMHCIIA) at the cell equator. Colocalization of staining of NMHCIIA and F-actin (phalloidin)-focusing zone (green: NMHCIIA, red: F-actin, blue: DAPI). (**B**) *Pafah1b1*[hc/ko] MEFs displayed dispersed and less-focused actomyosin at the cell equator.
*Figure 8 continued on next page*

*Figure 8 continued*

(green: NMHCIIA, red: F-actin, blue: DAPI). (**C**) Quantification of normal and dispersed NMHCIIA and F-actin distribution in MEFs. Scale bars: 10 μm. Quantitative data were included in *Figure 8—source data 1*.

The online version of this article includes the following source data for figure 8:

**Source data 1.** Quantification of MEFs.

LIS1 plays an important role in mitotic spindle orientation by controlling MTs and dynein complex in the interphase and metaphase of MEFs and neocortical NPCs (*Faulkner et al., 2000*; *Yingling et al., 2008*; *Moon et al., 2014*). However, during late stages of mitosis including cytokinesis, the cellular functions of LIS1 have not been clearly elucidated. Here we provide new evidence that LIS1 plays a crucial role in regulating cleavage plane positioning at late stages of mitosis and cytokinesis during early mouse development. We demonstrated the instructive functions of LIS1 in spindle orientation segregating the cell fate determinants into two daughter cells during mitosis, and LIS1 also regulates RhoA and Anillin-ring localization during cytokinesis. Collectively, this study provides key insights into underlying cellular mechanisms of which *Pafah1b1* deficiency leads to impairments in neurogenesis as well as defects in neuronal migration.

Since *Pafah1b1*$^{hc/ko}$ NPCs display severe mitotic progression perturbation causing cell cycle arrest at prometaphase and metaphase and frequent apoptosis in the neocortex (*Gambello et al., 2003*; *Yingling et al., 2008*), we needed to employ genetically manipulated MADM mouse lines to identify the sparsely labeled apical NPCs and daughter cells in *Pafah1b1* mutant embryonic brains in vivo. *Pafah1b1*$^{hc/ko}$ NPCs also exhibited severe mitotic defects and it was not possible to culture viable NPCs in vitro. Thus, it was unclear that *Pafah1b1* deficiency in NPCs leads to the same or similar cytokinetic phenotypes seen in MEFs in vitro. In the developing mouse brains, neocortical NPCs express specific nuclear transcription factors (such as PAX6, SOX2) to maintain its proliferative capacity and potency to promote neurogenesis. Neocortical NPCs undergo complex basal-to-apical abcission-mediated cytokinetic process in vivo and they also generate one daughter cell with a different fate, a neuron with different lineage depending on the birth date and subtypes of NPCs in regionally specified neocortical domain. By contrast, MEFs only generate two MEFs with symmetric divisions. Despite the different tissue origins of these two distinct cell types (MEFs and NPCs), both MEF and NPC populations share cellular and cytoarchitectural features during cytokinesis. We confirmed that NPCs retain apical RhoA and Anillin localization when they undergo cytokinesis at the ventricular surface where F-actin polarization occurred in vivo. RhoA and Anillin play crucial roles in determining cytokinetic furrow localization by cross-interacting with F-actin and Myosin II at the equatorial cell membrane and the midbody. It appears that RhoA and Anillin-mediated cytokinetic furrow formation is controlled in a similar manner in both MEFs in vitro and apical NPCs in vivo (vRGs). Multiple in vitro studies have highlighted that genetic mutant MEFs are versatile and reliable cell types to investigate cellular mechanisms underlying cytokinetic defects, including those in the developing brain (*Janisch et al., 2013*; *Rosario et al., 2010*; *Cook et al., 2011*; *Serres et al., 2012*). In genetically targeted MEFs with reduced protein dose (e.g. Kif20b, Plk4, Ect2, and P27), cytokinesis-associated phenotypes were found, including multi-nucleation and centrosome amplification. Therefore, the use of MEFs enabled us to perform long-term live-cell imaging for various analyses that were not possible in NPCs in vitro or in mouse brains in vivo.

These *Pafah1b1* mutant MEF studies of late mitosis and cytokinesis using live-cell imaging revealed that LIS1 restricts the cleavage furrow location at the equatorial cortex by inhibiting cell membrane hyper-contractility, demonstrating that LIS1 plays a critical role in cleavage plane specification as well as in actomyosin-mediated cell membrane contractility. We discovered that *Pafah1b1* mutant MEFs displayed abnormal cellular phenotypes at late stages of mitosis and cytokinesis, with increased incidence of binucleation and cell death resulted from the failure in cell separation. Cell shape oscillations and chromosome rocking along the mitotic spindles were also evident in *Pafah1b1* mutant MEFs, surprisingly similar to those of Anillin-depleted cells (*Echard et al., 2004*; *Kechad et al., 2012*; *Piekny and Glotzer, 2008*; *Straight et al., 2003*; *Zhao and Fang, 2005*). Together, these results suggest that *Pafah1b1* deficiency in MEFs induces cleavage furrow mispositioning and failure in the recruitment of contractile ring components to the equatorial cortex. Intriguingly, cell shape oscillation and cytokinesis defects were also previously reported in nocodazole-

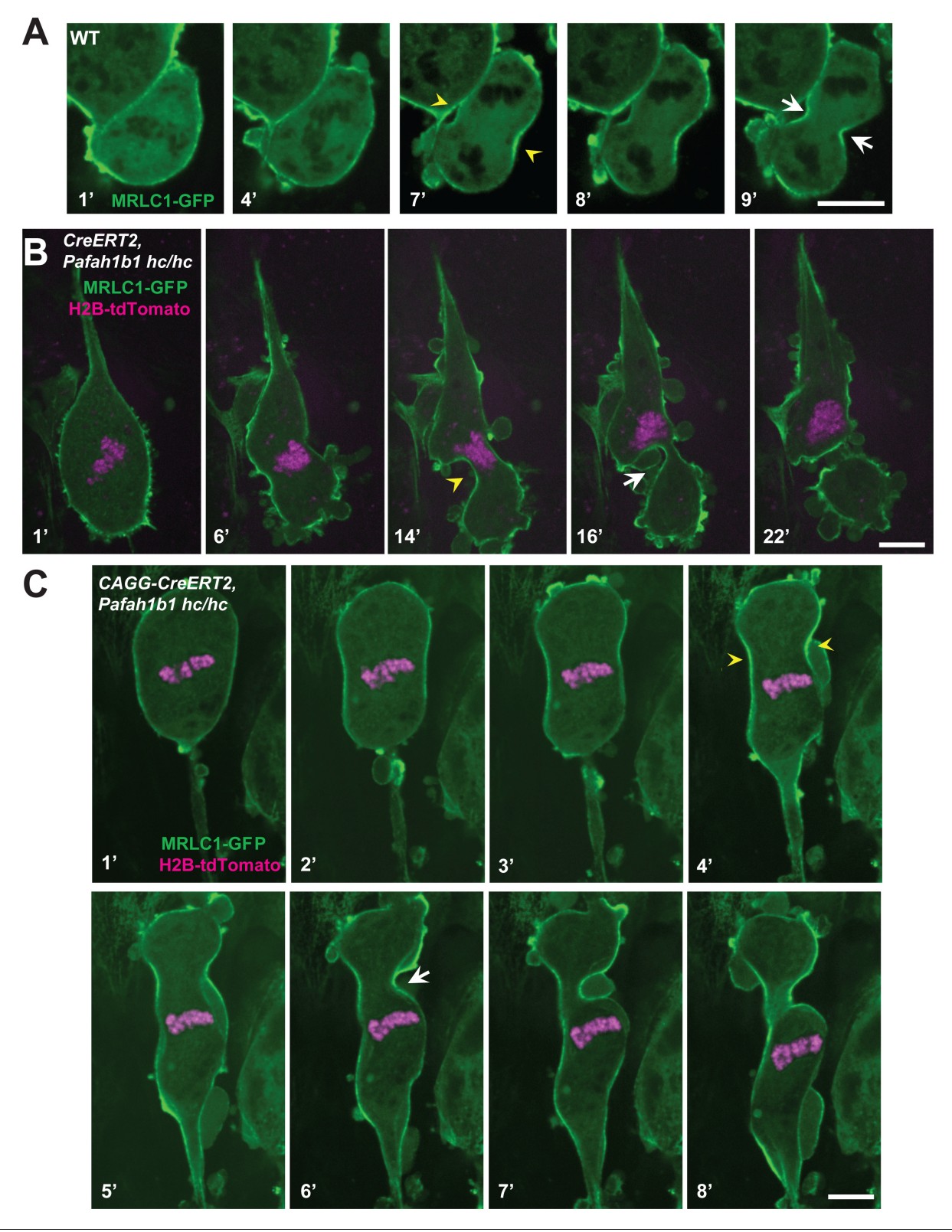

**Figure 9.** Myosin-II is abnormally distributed in *Pafah1b1* mutant MEFs during late stages of mitosis. (A) WT MEFs displayed normal recruitment of the myosin regulatory light chain 1 (MRLC, green) of Myosin-II subunit at the equatorial cortex during cytokinesis. Dark black spots inside of the cells indicate chromosome sets. (B) *Pafah1b1* mutant MEFs (*CAGG-CreERT2; Pafah1b1^hc/hc^* + TM^24h^) infected with MRLC1-GFP (green) and H2B-tdTomato (magenta), displayed cytokinesis failure with an asymmetrically mispositioned cleavage furrow. (C) *Pafah1b1* mutant MEFs clearly displayed uncoupling

*Figure 9 continued on next page*

*Figure 9 continued*

between chromosome separation (usually before cytokinesis onset) and cleavage furrow ingression. *Pafah1b1* mutant MEFs (4') still maintained unsegregated tetraploid chromosomes to initiate furrow contraction from the actomyosin ring. (Yellow arrowheads: initial accumulation of MRLC1, White arrows: final cleavage furrow positioning). Scale bars: 5 µm.

treated cells (*Canman et al., 2003*), suggesting that misregulation of either MT or the MT-actomyosin interface at the cell membrane may induce hyper-contractility leading to abnormal membrane contraction in *Pafah1b1* mutant MEFs. During cytokinesis, *Pafah1b1*-dependent cell membrane contractility may be mediated indirectly by the dyregulation of astral MTs reaching to the cell membrane (*Moon et al., 2014*).

We aimed to identify the molecular mechanisms underlying *Pafah1b1*-dependent cell membrane contractility of MEFs. Therefore, we examined the distribution of the other proteins defining cleavage furrow formation including RhoA and its downstream effectors, like contractile ring components and actomyosin. We found that RhoA and Anillin were mislocalized in *Pafah1b1* mutant MEFs. For instance, RhoA was found in ectopically localized in large polar blebs in *Pafah1b1* mutant MEFs. The Myosin II-positive and F-actin focus zone were dispersed at the equatorial cortex or aberrantly located at the polar blebs in *Pafah1b1* mutant MEFs. By alternatively tracing MRLC1 and SEPT6, we confirmed that LIS1 defines cleavage plane positioning by dual regulation of actomyosin and contractile ring. Finally, RhoA hyper-activation in WT MEFs phenocopied the cytokinesis defects seen in *Pafah1b1* mutant MEFs, and inhibition of RhoA in *Pafah1b1* mutant MEFs decreased the cytokinesis defects observed in these MEFs. Thus, these results suggest that LIS1 exerts its function to restrict the cleavage furrow at the equatorial cortex to finely modulate RhoA-Anillin-actomyosin signaling.

Previous *Pafah1b1* mutant mouse studies demonstrated that LIS1 modulates GTPase-mediated F-actin dynamics in interphase cells and post-mitotic neurons. For example, *Pafah1b1* heterozygous (*Pafah1b1^{ko/+}*) post-mitotic neurons displayed F-actin dysregulation at the leading processes and aberrant GTPase activities of RhoA-Rac1-Cdc42. Similarly, *Pafah1b1^{ko/+}* MEFs exhibited abnormal hyper-activation of RhoA GTPase (*Kholmanskikh et al., 2003*). A recent *Pafah1b1* knockdown fibroblast study also unraveled F-actin dysfunction and focal adhesion disruption during traction-dependent migration (*Jheng et al., 2018*). Together, these previous findings indicate that LIS1 may regulate not only MTs but F-actin in migrating phases of multiple cell types. Based on the aberrant cytokinetic phenotypes seen in *Pafah1b1* mutant NPCs and MEFs, we now demonstrate that loss of LIS1 results in hyper-activation of RhoA signaling and mislocalization of F-actin and Myosin II that generates hyper-contractility forces ectopically at the polar cortex, which is an abnormal site of cleavage furrow ingression.

Multiple mouse studies have suggested that RhoA GTPase regulates the integrity of the apical NPC niche, through controlling F-actin at the adherens junction (*Cappello et al., 2012*; *Herzog et al., 2011*; *Katayama et al., 2011*). In accordance with this finding, a study of chick neural tube development indicated that RhoA activity modulation and overexpression of a DN-RhoA similarly altered mitotic spindle orientation of apical NPCs (*Roszko et al., 2006*). In the present study, we demonstrated that neocortical LIS1 is an important upstream mediator of signaling cascades of RhoA-actomyosin-contractile ring components during cytokinesis of apical

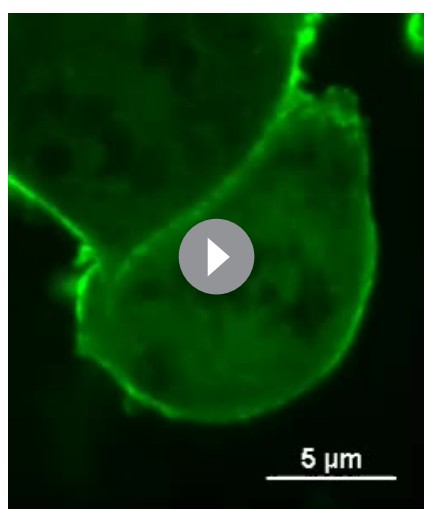

**Video 3.** Myosin II localization during cytokinesis of wild-type (WT) MEFs. Time-lapse live cell imaging of mitotic cell division from wild-type (WT) MEFs. Myosin regulatory light chain 1 (MRLC1)-GFP and H2B-tdTomato labeled fluorescence signals were acquired with a 30 s interval by Nikon Ti spinning disk confocal microscope. In normal cytokinesis, MRLC1 was recruited to the equatorial cortex and formed the cleavage furrow.

https://elifesciences.org/articles/51512#video3

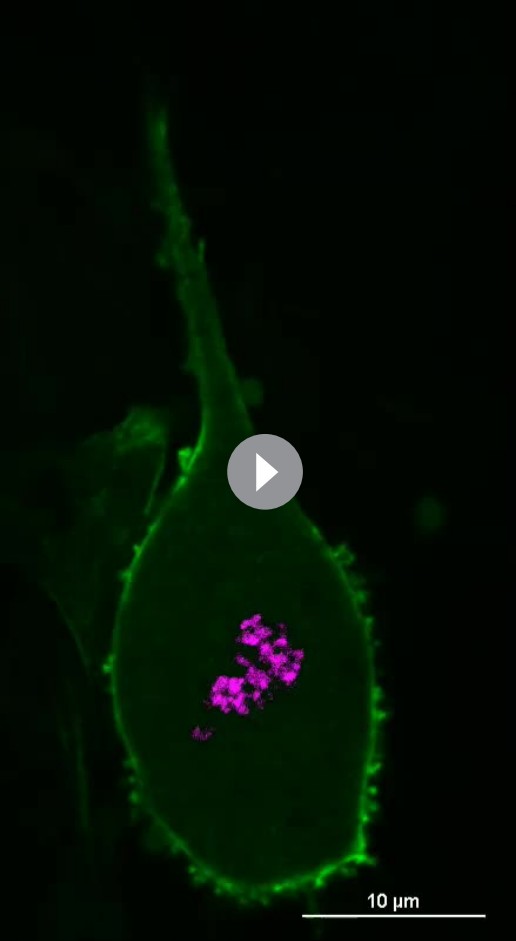

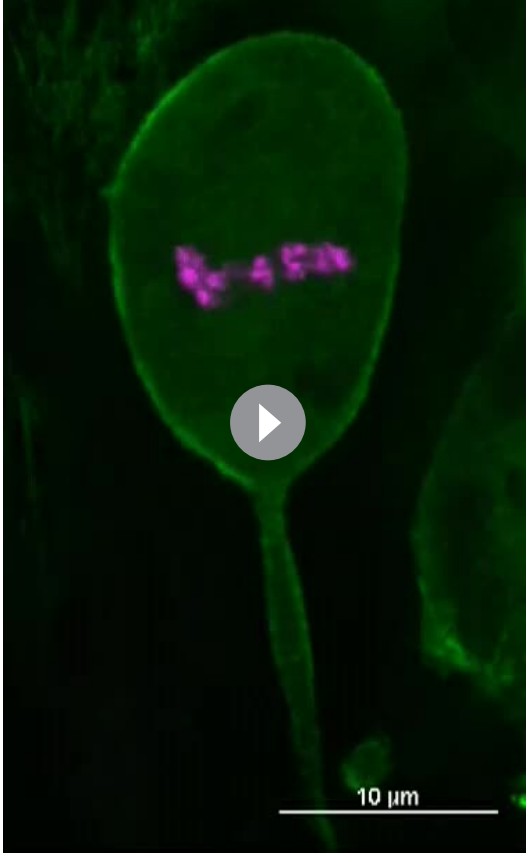

**Video 4.** Abnormal Myosin II movements during cytokinesis of *Pafah1b1* mutant MEFs. Time-lapse live cell imaging of mitotic cell division from *Pafah1b1* mutant MEFs (*CAGG-CreERT2;Pafah1b1^{hc/hc}* +TM^{24 h}). Myosin regulatory light chain 1 (MRLC1)-GFP and H2B-tdTomato labeled fluorescence signals were acquired with a 30 s interval by Nikon Ti spinning disk confocal microscope. In *Pafah1b1* mutant MEFs, MRLC1 was first recruited to the equatorial cortex. However, we found failure to properly restrict the cleavage furrow at the equatorial cortex.
https://elifesciences.org/articles/51512#video4

**Video 5.** Uncoupling between chromosome segregation and cytokinesis in *Pafah1b1* mutant MEFs. Time-lapse live cell imaging of mitotic cell division from *Pafah1b1* mutant MEFs (*CAGG-CreERT2; Pafah1b1^{hc/hc}* +TM^{24 h}). Myosin regulatory light chain 1 (MRLC1)-GFP and H2B-tdTomato labeled fluorescence signals were acquired with a 30 s interval by Nikon Ti spinning disk confocal microscope. We observed frequent uncoupling between chromosome segregation and cytokinesis in *Pafah1b1* mutant MEFs.
https://elifesciences.org/articles/51512#video5

NPCs. In mouse apical NPCs, Anillin-ring constriction from the basal to apical side determines the cleavage furrow ingression site in mouse apical NPCs (*Kosodo et al., 2008*). During cytokinesis of neocortical NPCs, Citron kinase and RhoA colocalize at the apical membrane of NPCs where the cleavage furrow and the midbody are positioned (*Di Cunto et al., 2000*; *Sarkisian et al., 2002*). In mice, mutations in Citron kinase alter mitotic spindles and RhoA-mediated neurogenic cytokinesis of apical NPCs. In humans, neocortical NPC phenotypes lead to primary microcephaly and microlissencephaly (*Basit et al., 2016*; *Li et al., 2016*; *Shaheen et al., 2016*; *Harding et al., 2016*; *Gai and Di Cunto, 2017*). Furthermore, in addition to Citron kinase and RhoA, three members of Kinesin family, Kif20a, Kif20b, and Kif4 regulate neocortical NPC cytokinesis and mutations of these genes result in reduced cortical size with microcephaly (*Janisch et al., 2013*; *Moawia et al., 2017*; *Geng et al., 2018*). Together, these previous studies suggest that dynamic movements of RhoA and cleavage furrow-associated proteins to the midbody is the key cellular process regulating NPC cytokinesis by precisely segregating cell determinants along the cell cleavage axis.

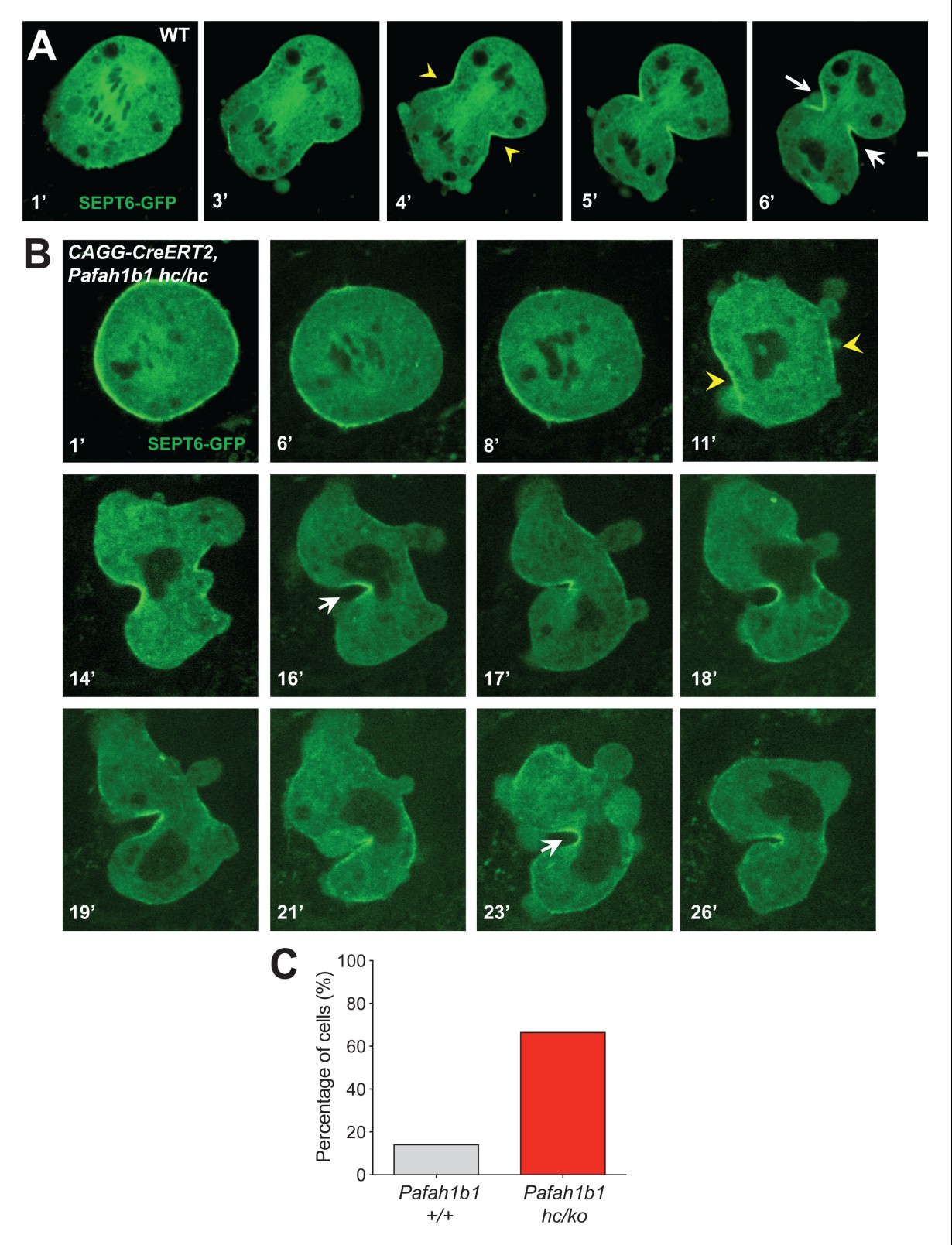

**Figure 10.** The contractile ring component SEPT6 is not properly maintained at the equatorial cortex in *Pafah1b1* mutant MEFs. (**A**) WT MEFs displayed normal recruitment of Septin 6 (SEPT6-GFP, in green), an actomyosin-fiber crosslinking protein composed of contractile ring, to the equatorial cortex. (**B**) *Pafah1b1* mutant MEFs (*CAGG-CreERT2; Pafah1b1^hc/hc^ + TM^24h^*) displayed cell shape oscillation with spindle/chromosome rocking. *Pafah1b1* mutant MEFs displayed chromosome missegregation and failure to maintain proper contractile ring contraction sites. Dark black

*Figure 10 continued on next page*

*Figure 10 continued*

spots inside of the cells indicate the chromosome sets. (Yellow arrowheads: initial accumulation of SEPT6, White arrows: final cleavage furrow positioning). (**C**) Quantification of cytokinetic failure with abnormal SEPT6-GFP distribution indicates that binucleation events and incomplete cytokinesis occur more frequently in *Pafah1b1*-deficient mutant MEFs than WT control MEFs. Scale bars: 10 μm. Quantitative data was included in *Figure 10—source data 1*.

The online version of this article includes the following source data for figure 10:

**Source data 1.** Quantification of MEFs.

In the developing neocortex, apical NPCs self-renew via symmetric divisions, and produce the major populations of post-mitotic neurons via asymmetric divisions. The orientation of the cleavage plane is thought to be a major regulator of symmetric versus asymmetric cleavage. Our present study suggests that the LIS1-RhoA-actomyosin pathway sets the apical NPC cleavage plane by enrichment of cytokinetic furrow proteins to the ventricular surface, contributing to the later mitotic events when cell fate determinants separate into the daughter cells. Immunohistochemical analyses of the mouse neocortex uncovered that *Pafah1b1* mutant NPCs displayed mislocalization of RhoA and Anillin skewed to only one daughter cell, indicating unequal inheritance and this may impact the segregation of cell fate determinants by inducing an imbalance of symmetric versus asymmetric divisions. Consistent with this data, we previously reported that *Pafah1b1*-deficient E14.5 neocortex had an increase in TUJ1-positive neurons and cleaved caspase-3-positive apoptotic cells (*Gambello et al., 2003*). This suggests that neocortical daughter cell fates and survival/death are sensitive to LIS1 expression levels in neocortical NPCs. We also demonstrated that *Pafah1b1*-deficient E15.5 neocortex displayed partial diffuseness and thinning of cortical plate with weak formation of the subplate. In the cortex of postnatal day 5 NPC-specific *Pafah1b1* CKO mice (*GFAP-Cre;*

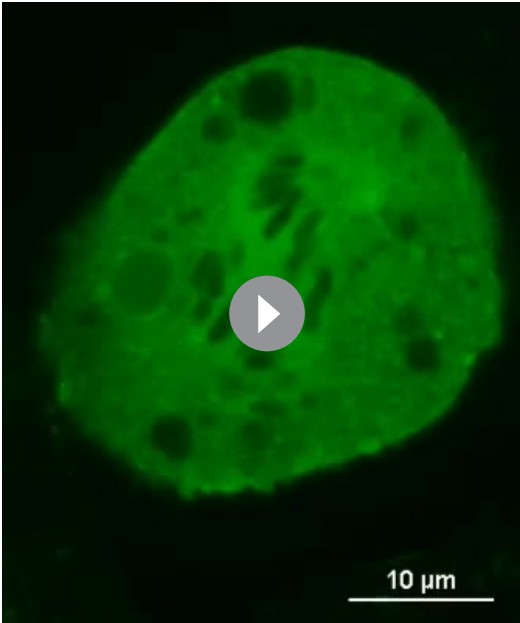

**Video 6.** Septin localization during cytokinesis of wild-type (WT) MEFs. Time-lapse live cell imaging of mitotic cell division from wild-type (WT) MEFs. Septin 6 (SEPT6)-GFP labeled fluorescence signals were acquired with a 30 s interval by Nikon Ti spinning disk confocal microscope. In normal cytokinesis, Septin-associated contractile ring complex was recruited to the equatorial cortex and formed the cleavage furrow.

https://elifesciences.org/articles/51512#video6

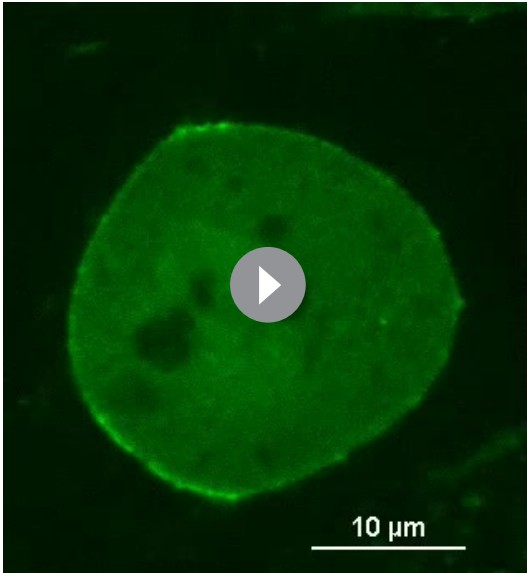

**Video 7.** Abnormal Septin localization during cytokinesis of *Pafah1b1* mutant MEFs. Time-lapse live cell imaging of mitotic cell division from *Lis1* mutant MEFs (*CAGG-CreERT2; Pafah1b1*$^{hc/hc}$ +TM$^{24\ h}$). Septin 6 (SEPT6)-GFP labeled fluorescence signals were acquired with a 30 s interval by Nikon Ti spinning disk confocal microscope. We observed SEPT signals at the equatorial cortex initially but then it regressed with vigorous cortical deformation and chromosome oscillation/rocking.

https://elifesciences.org/articles/51512#video7

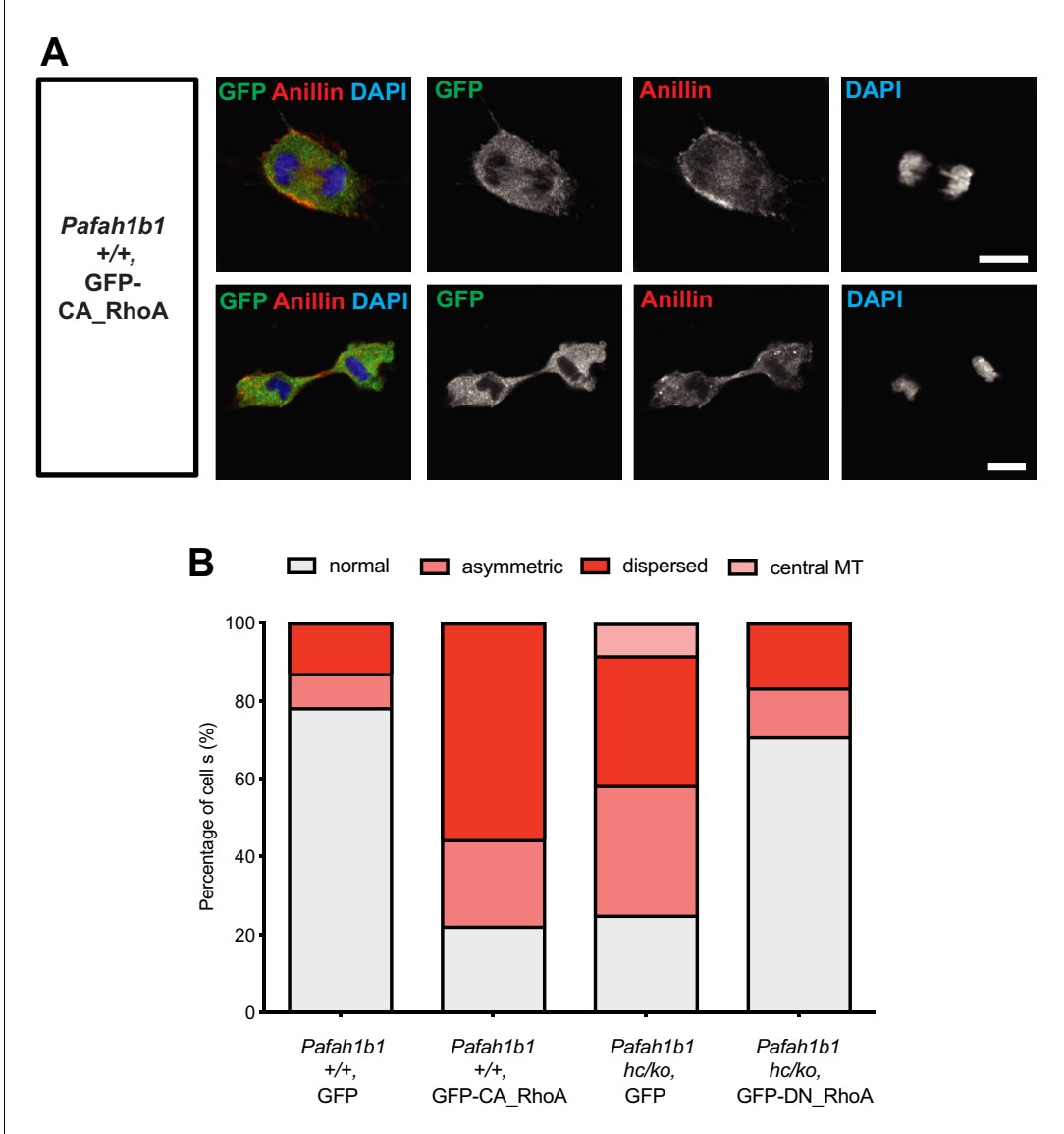

**Figure 11.** Overexpression of a constitutively active form of RhoA recapitulates cytokinetic failure similar to those are seen *Pafah1b1* mutant MEF and a dominant negative form or RhoA rescues hyper-activity of RhoA observed in *Pafah1b1* mutant MEFs. (A) WT MEFs expressing a constitutively active form of RhoA (CA-RhoA) displayed abnormal Anillin distribution during cytokinesis consistent with cytokinesis defects seen in *Pafah1b1^{hc/ko}* MEFs. (B) The *Pafah1b1^{hc/ko}* MEFs expressing a dominant negative form of RhoA (DN-RhoA) displayed partially rescued cytokinesis phenotypes similar to WT control MEFs. Scale bars: 10 μm. Quantitative data were included in *Figure 11—source data 1*.

The online version of this article includes the following source data for figure 11:

**Source data 1.** Quantification of MEFs.

*Pafah1b1^{hc/hc}*) also had reduced numbers of upper-layer neurons marked by FOXP1 (Layer III/VI) and CUX1 (Layer II/III) (*Youn et al., 2009*), indicating that *Pafah1b1* deficiency in NPCs leads to impairments in neuronal subtype specification and total neurogenesis accompanied with cortical patterning defects.

In the present study, we observed mitotic phenotypes from apical NPCs, the vRGs, in the E14 neocortex. *Pafah1b1* deficiency at earlier time points (E9.5∼E11.5) resulted in neural epithelial stem cell death via apoptosis, due to disruption of symmetric mitotic division with vertical cleavage plane, so these earlier stem cells could not be studied. At mid-gestation, we were able to utilize conventional hypomorphic mutants and NPC-specific CKO mutants and circumbented mitotic arrest and

death mechanisms. E14 vRGs had a sufficient amount of LIS1 to avoid severe mitotic catastrophe and still displayed robust cytokinetic defects with distorted cleavage plane and mitotic spindles.

Our novel findings of LIS1 functions in cytokinesis may explain the mechanism of the recent studies with induced pluripotent stem cells (iPSCs) of Miller Dieker syndrome, a severe form of lissencephaly with haploinsufficiency of *PAFAH1B1* as well as about 20 other genes. We demonstrated a prolongation of mitosis of oRG progenitors but not ventricular zone radial glial (vRG) progenitors (*Bershteyn et al., 2017*). The specificity of mitotic effects in the oRGs and lack of its effects in the vRGs may be the result of different dose dependencies of LIS1 in mouse vs. human, such that LIS1 deficiency results in more prolonged mitotic time specifically in oRGs rather than vRGs. Compared with vRGs, oRG populations lack their endfeet attachment to the apical VZ surface. Intriguingly, human oRGs exhibit rapid mitotic somal translocation toward the cortical plate right before cytokinesis (*Hansen et al., 2010*). These abrupt movements of human oRGs suggest dynamic cytoskeletal rearrangements during mitotic phases of oRGs. In our current mouse studies, we could not investigate oRG phenotypes due to the low number oRGs present in the neocortices of mice. By contrast, the neocortices of gyrencephalic mammals (from the ferret to primates) contain substantial number of oRGs that exhibit unique mitotic somal translocation, unrestricted mitotic spindle angles, multiple rounds NPC self-renewal and promote expansion of progenitors, neocortical folding and increased topology. Since oRGs highly express a specialized transcriptional factor, HOPX (*Pollen et al., 2015*) on top of the pan-RG markers PAX6 and SOX2, it would be crucial to investigate the orchestrated crosstalk between HOPX-dependent transcriptional/epigenetic network and actomyosin-mediated cytokinesis in oRGs. In the future, characterization of the localization of diverse protein markers enriched to the cleavage furrow in human or primate iPSC-derived NPCs and fetal neocortical NPCs, such as RhoA, Anillin, and Citron kinase of NPCs, will need to be analyzed to better understand the oRG-specific sensitivity to LIS1 dosage especially in human. It would also be interesting to determine whether other lissencephaly or brain malformation-associated genes modulate RhoA-Anillin-actomyosin-pathways, and to test whether these genes, like LIS1, also possess dual roles in actomyosin and MT regulation important for apical NPCs by controlling key cellular processes not only mitosis but cytokinesis, a critical final step of cell determinant segregation.

# Materials and methods

## Key resources table

| Reagent type (species) or resource | Designation | Source or reference | Identifiers | Additional information |
|---|---|---|---|---|
| Strain, strain background *Mus musculus* | *CAGG-CreERT2* | *Hayashi and McMahon, 2002* | | |
| Strain, strain background *Mus musculus* | *Pafah1b1$^{hc/+}$* *Pafah1b1$^{hc/hc}$* *Pafah1b1$^{hc/ko}$* | *Yingling et al., 2008*; *Moon et al., 2014* | | |
| Strain, strain background *Mus musculus* | *GFAP-Cre* | *Zhuo et al., 2001* | | |
| Strain, strain Background *Mus musculus* | *MADM-11$^{GT/TG}$; Emx1$^{Cre+}$* | *Hippenmeyer et al., 2010* | | |
| Cell line (*Homo sapiens*) | HEK293T (kidney, embryonic) | ATCC | CRL-3216 | Negative for Mycoplasma contamination |
| Antibody | Anti-Cleaved caspase3 (Rabbit polyclonal) | Cell Signaling | Cat# 9661 (RRID:AB_2341188) | ICC (1:500) |
| Antibody | Anti-α-Tubulin (Rat monoclonal) | AbD Serotec | Cat# MCA77G (RRID:AB_325003) | ICC (1:1,000) |

*Continued on next page*

*Continued*

| Reagent type (species) or resource | Designation | Source or reference | Identifiers | Additional information |
|---|---|---|---|---|
| Antibody | Anti-RhoA (Mouse monoclonal) | Santa Cruz | Cat# sc-418 (RRID:AB_628218) | ICC, IHC (1:500) |
| Antibody | Anti-Anillin (Rabbit polyclonal) | Santa Cruz | Cat# sc-67327 (RRID:AB_2058302) | ICC, IHC (1:250) |
| Antibody | Anti-LIS1 (Rabbit polycloncal) | Abcam | Cat# ab2607 (RRID:AB_2299251) | ICC (1:250) |
| Antibody | Anti-P150$^{glued}$ (Mouse monoclonal) | BD bioscience | Cat# 610473 (RRID:AB_397845) | ICC (1:200) |
| Antibody | Anti-DIC 74.1 (Mouse monoclonal) | Millipore | Cat# MAB1618 (RRID:AB_2246059) | ICC (1:100) |
| Antibody | Anti-NMHCIIA (Rabbit polyclonal) | Covance | Cat# PRB-440P (RRID:AB_291638) | ICC (1:1,000) |
| Antibody | Anti-GFP (Mouse monoclonal) | Invitrogen | Cat# A-11120 (Clone 3E6) (RRID:AB_221568) | ICC (1:500) |
| Antibody | Anti-MPM2 (Mouse monoclonal) | Millipore | Cat# 05–368 (RRID:AB_309698) | IHC (1:200) |
| Antibody | Anti-N-Cadherin (Mouse monoclonal) | BD bioscience | Cat# 610920 (RRID:AB_2077527) | IHC (1:100) |
| Antibody | Anti-aPKC (PKCζ) (Rabbit polyclonal) | Santa Cruz | Cat# sc-216 (Clone C-20) (RRID:AB_2300359) | IHC (1:100) |
| Antibody | Anti-GFP (Chicken polyclonal) | Aves | Cat# GFP1010 (RRID:AB_2307313) | IHC (1:500) |
| Antibody | Anti-c-Myc (Goat polyclonal) (detecting tdTomato-c-Myc) | Novus biologicals | Cat# NB-600–335 (RRID:AB_10002720) | IHC (1:150) |
| Recombinant DNA reagent | pCX-mCherry-αTub | Previously generated. *Moon et al., 2014* | | |
| Recombinant DNA reagent | pCX-MRLC1-GFP | Tom Egelhoff (Cleveland Clinic, Cleveland, OH, USA) | | Addgene plasmid #35680 |
| Recombinant DNA reagent | pCX-SEPT6-GFP | Matthew Krummel (UCSF, San Francisco, CA, USA) | | Described in *Gilden et al. (2012)* |
| Recombinant DNA reagent | pCX-H2B-tdTomato | Geoffrey Wahl (Salk Institute, San Diego, CA, USA) Roger Tsien (UCSD, San Diego, CA, USA) | | Addgene plasmid #17735 |
| Recombinant DNA reagent | pCX-DN-RhoA, pCX-CA-RhoA | Kozo Kaibuchi (Nagoya University, Nagoya, Japan) | | |

## Animals

The mouse lines used in the present study were previously described; *CAGG-CreERT2* (*Hayashi and McMahon, 2002*), *Pafah1b1$^{hc/+}$*,*Pafah1b1$^{hc/hc}$* and *Pafah1b1$^{hc/ko}$* (*Yingling et al., 2008*; *Moon et al.,*

2014), *GFAP-Cre* (*Zhuo et al., 2001*), *MADM-11^{GT/TG,Lis1};Emx1^{Cre/+}* (*Hippenmeyer et al., 2010*). All animal care and experimental procedures (Protocol ID: AN085137-02A) were approved by the University of California, San Francisco Institutional Animal Care and Use Committee (IACUC) in accordance with the National Institutes of Health Guide for the Care and Use of Laboratory Animals.

## Cell culture and immunocytochemistry

Primary MEFs cell culture was performed as previously described *Moon et al. (2014)*. Cre recombinase in *CAGG-CreERT2*-MEFs was induced by administration of 4-hydroxy tamoxifen (TM) (Sigma, 100 nM) for 12 hr. DMSO and blebbistatin (Calbiochem, 100 µM) were used to treat MEFs for 30 min. To identify RhoA in the cortex and cleavage furrow, MEFs were fixed with freshly made 10% trichloroacetic acid (TCA) for 10 min (*Yonemura et al., 2004*). Immunocytochemistry staining of cortical P150^{glued}, was performed in 100% methanol-fixed samples as previously described *Moon et al. (2014)*. The antibodies used in this study were listed in Key Resources Table.

## Retrovirus infection, epifluorescence and spinning-disk confocal live-cell imaging

Time-lapse live-cell imaging of MEFs infected with retroviruses encoding H2B-GFP, mCherry-α-Tubulin was described previously *Moon et al. (2014)*. To induce Cre-mediated *Pafah1b1* CKO allele deletion, the MEFs derived from *CAGG-CreERT2; Pafah1b1^{+/+}* and *CAGG-CreERT2; Pafah1b1^{hc/hc}* animals were treated with 4-hydroxy-tamoxifen (TM) (Sigma, 100 nM) for 12 hr or 24 hr. To visualize Myosin II movements during cytokinesis, primary MEFs were co-infected with a mixture of MRLC1-GFP and H2B-tdTomato retroviruses with 4 µg/mL polybrene. Lasers with 488 nm and 561 nm emission were used for GFP and tdTomato imaging. SEPT6-GFP was also traced by a Nikon Ti spinning-disk confocal microscope with a 488 nm emission laser. The sequential 2 µm interval 5 ~ 7 Z-stack images were captured every 30 s. The best Z confocal plane images showing the cleavage furrow were made as time-lapse movies. The plasmid sources of retrovirus construction were described in Key Resources Table.

## Immunohistochemistry

Mouse fetal brains were isolated from E14.5 embryos described in *Yingling et al. (2008)* followed by fixation with 4% paraformaldehyde (PFA) or 10% TCA (RhoA staining). On the next day, the brains were transferred to 30% sucrose and embedded in O.C.T. compound to make cryomolds. 18 µm- thick cryostat sections of neocortex were used for immunohistochemistry. In *MADM-11* animals, we identified *Lis1* genotype based on GFP or tdT immunoreactivity. Cleavage furrow markers such as Anillin and RhoA were immunostained in the far-red Alexa Fluor 647 channel to separate the fluorescence signal from GFP and tdT. Anillin- or RhoA-immunoreactive signals were pseudo-colored in green for representation purpose. The antibodies used in this study were listed in Key Resources Table.

## Sample size, Replicates, and Statistical analysis

Time-lapse live-cell imaging of primary MEFs was performed and data analyses were conducted from at least three independent experimental sets. Sample-size from live-cell imaging for data analysis was calculated based on empirical sample-size previously tested from mitotic analysis of *Pafah1b1* mutant MEF study (*Moon et al., 2014*). In all MEF live-cell imaging experiments, the exact cell number recorded for monitoring whole duration of mitotic cell division (from prometaphase to telophase/cytokinesis) were written in Source data files. From the fixed samples for immunohistochemistry using the embryonic brains, the exact number of cells taken for confocal images of apical NPCs located at the ventricular zone was recorded in Source data files. In immunocytochemistry analysis of MEFs, the exact sample number of cells for taking confocal images was recorded in Source data files. No data points were excluded for statistical analysis and all raw data points were included. Two-tailed Student's *t*-test was used to compare mitotic cellular phenotypes and to evaluate statistical significance and differences between controls and *Pafah1b1* mutant groups. *p<0.05, **p<0.01, ***p<0.001, ns: not significant. Bars in the graphs with error bars represented: mean ± S.E.M.

## Acknowledgements

We would like to thank Dr. Arshad Desai and Dr. Karen Oegema in UCSD for providing initial key insights into the novel phenotypes seen in *Lis1* mutant MEFs, Dr. Torsten Wittmann, Hayley Pemble, Kurt Thorn at Nikon imaging center in UCSF for access to microscopes for time-lapse live-cell imaging, Dr. Matthew Krummel in UCSF for Septin6-GFP plasmid, Dr. Kozo Kaibuchi in Nagoya University for RhoA plasmids and insightful advice, and Dr. Giles Plant in Stanford University for access to Nikon imaging analysis.

## Additional information

### Funding

| Funder | Grant reference number | Author |
|---|---|---|
| National Institute of Neurological Disorders and Stroke | NIH-R01-NS041030 | Anthony Wynshaw-Boris |
| Eunice Kennedy Shriver National Institute of Child Health and Human Development | NIH-R01-HD047380 | Anthony Wynshaw-Boris |
| University of California, San Francisco | Graduate Student Research Award | Hyang Mi Moon |

The funders had no role in study design, data collection and interpretation, or the decision to submit the work for publication.

### Author contributions

Hyang Mi Moon, Conceptualization, Resources, Data curation, Formal analysis, Supervision, Validation, Investigation, Visualization, Methodology, Project administration; Simon Hippenmeyer, Resources, Visualization, Methodology; Liqun Luo, Resources, Methodology; Anthony Wynshaw-Boris, Conceptualization, Data curation, Supervision, Funding acquisition, Validation, Investigation, Project administration

### Author ORCIDs

Hyang Mi Moon https://orcid.org/0000-0003-2755-3767
Simon Hippenmeyer http://orcid.org/0000-0003-2279-1061
Liqun Luo http://orcid.org/0000-0001-5467-9264
Anthony Wynshaw-Boris https://orcid.org/0000-0002-2780-1540

### Ethics

Animal experimentation: All animal care and experimental procedures were approved by the University of California, San Francisco Institutional Animal Care and Use Committee (IACUC) in accordance with the National Institutes of Health Guide for the Care and Use of Laboratory Animals. Protocol ID: AN085137-02A.

### Decision letter and Author response

Decision letter https://doi.org/10.7554/eLife.51512.sa1
Author response https://doi.org/10.7554/eLife.51512.sa2

## Additional files

### Supplementary files

• Transparent reporting form

### Data availability

All data analyzed during this study and its analysis has been described in the manuscript.

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
