## [Decision Letter]

**Acceptance summary:**

The authors carry out detailed experiments aimed at uncovering the role of LIS1-Dynein-Microtubules in cytokinetic plane specification. They investigate the functions in two cell types, Neural Progenitor Cells and Mouse Embryonic Fibroblasts. Through extensive genetic and imaging studies the authors reveal links between LIS11 and the RhoA-Anillin-Actin-Myosin II and how this network positions the cell division site.

**Decision letter after peer review:**

Thank you for submitting your article "LIS1 determines cleavage plane positioning by regulating actomyosin-mediated cell membrane contractility" for consideration by *eLife*. Your article has been reviewed by three peer reviewers, and the evaluation has been overseen by a Reviewing Editor and Anna Akhmanova as the Senior Editor. The following individual involved in review of your submission has agreed to reveal their identity: Hongyan Wang (Reviewer #1).

The reviewers have discussed the reviews with one another and the Reviewing Editor has drafted this decision to help you prepare a revised submission.

All the three reviewers were enthusiastic about the new role for LIS1 that you and colleagues describe in actomyosin and membrane contractility in neocortical neural progenitor cells. They have raised a number of points, which have all been compiled below and I request you to consider them carefully. One point brought up more than once by the reviewers is the modest quantitative information in your paper. This needs to be improved. Please pay particular attention to the points raised below, with respect to the physical interactions, cell type specificity, as well as relevance of MEF work. A number of new experiments may be required to address these points, in addition to substantial rewriting as suggested in point 6.

Essential revisions:

1) Does LIS1 physically associates with the RhoA and Anillin to regulate cytokinesis? Or the regulation between LIS1 and RhoA and Anillin is indirect. Please explain these possibilities.

2) Quantitative data need to be improved. Please indicate the exact sample size of experiment whenever possible. Very often the sample sizes in the manuscript are shown as "more than twenty cells", "more than twenty apical NPCs", "at least thirty cells". The authors should indicate the n number in each experiment and figure and present the data as "mean ± S.E.M" as much as possible. The population of abnormal division of NPCs were not quantified in Figure 1 and Figure 2.

3) It would be helpful to connect the disordered division, and shifts in asymmetric and symmetric division that the authors report to the ultimate functional impact in the neuronal cell population. Further could the authors clarify if this dependence is somehow specific to this subtype of neuronal precursors?

4) How much of the data generated in the MEFs is relevant for the NPCs of the neocortex? Is it possible to isolate neuronal precursors from the Lis1 null mice, or differentiate neurons towards these lineage to validate that some of the findings in MEFs hold true for NPCs? For example, is it possible to check whether the separation defects so effectively demonstrated in MEFs actually also occur in NPCs?

5) The paper risks becoming somewhat descriptive at times. Better integration of the findings especially those in the MEFs, and connecting them to how Lis1 may impact NPC generation and functionality could help resolve this.

6) While the results are well documented, the text needs considerable improvement. The authors did raise interesting questions in the Introduction, for example: why do LIS1 mutants specifically display a decrease in neuroepithelial stem cells in the neocortex compared with a less catastrophic phenotype seen in radial glial (RG) progenitors; does LIS1 have differential roles in mitotic division time between ventricular and subventricular radial glial progenitors? However, there is little answer to these questions in the paper with the current experimental design and methods. The work was partly carried out on mouse progenitor cells without differentiation of epithelial stem cell and radial glia cells and partly on MEFs which have significant limitations relevant to answer the main the questions. The difference between neuronal migration and cortical development in lisencephalic rodent (mouse) and gyrensephlaic primate (human) is not sufficiently emphasized. The Introduction and Discussion are not scholarly written and historically informed to be useful to a multidisciplinary audience of a journal like *eLife*. For example, while citing themselves profusely (10 times!?), the seminal papers by Reiner Orly on LIS1 gene are conspicuously missing.

---

## [Author Response]

All the three reviewers were enthusiastic about the new role for LIS1 that you and colleagues describe in actomyosin and membrane contractility in neocortical neural progenitor cells. They have raised a number of points, which have all been compiled below and I request you to consider them carefully. One point brought up more than once by the reviewers is the modest quantitative information in your paper. This needs to be improved. Please pay particular attention to the points raised below, with respect to the physical interactions, cell type specificity, as well as relevance of MEF work. A number of new experiments may be required to address these points, in addition to substantial rewriting as suggested in point 6.

Please see our responses to Essential revisions #1, #3 and #5.

Essential revisions:1) Does LIS1 physically associates with the RhoA and Anillin to regulate cytokinesis? Or the regulation between LIS1 and RhoA and Anillin is indirect. Please explain these possibilities.

Physical interactions.

We thank the reviewers raising these interesting points regarding physical interaction of LIS1 and cytokinesis regulators (RhoA and Anillin). Although we performed immunocytochemistry experiments on MEFs with the individual antibodies during later stage of mitosis and cytokinesis, it was not possible to perform the co-localization experiments between LIS1 and other cytokinesis regulators. Since LIS1 and Anillin antibodies are derived from the same species, Rabbit, co-localization between these two proteins (LIS1-Anillin) was not able to be examined on MEFs simultaneously. On the other hand, membrane-bound RhoA required specific fixation method (TCA solution). However, endogenous intracellular distribution of LIS1 was different in 4% PFA vs. 10% TCA-fixation. Here, in the response to reviewers, we attached immunocytochemistry images of antibodies detecting LIS1 and Anillin using MEFs. We found no evidence of substantial overlap of intercellular compartments of LIS1-Anillin. In mitotic division of wild-type (WT) MEFs, LIS1 is localized only to the centrosomes in metaphase-to-anaphase and the central MTs during cytokinesis displaying mostly cytosolic distribution (Figure 7A), while Anillin is initially seen on the lateral cortex and later specifically accumulate in the cytokinetic abcission ring between two central MT bundles, near the equatorial cortex (Figure 7B). As we have shown in WT MEFs (Figure 6A), RhoA and Anillin are co-localized at the equatorial cortex and the midbody during cytokinesis. However, these two proteins display distinct membrane-associated enrichment compared with cytosolic LIS1. Our experimental evidence supports that LIS1 may only ‘indirectly’ regulate RhoA and Anillin through F-actin cytoskeleton regulation. A previous study (Kholmanskiki et al., 2003) demonstrated that *Pafah1b1*-deficient heterozygous MEFs have less amounts of the active form of GTP-bound RhoA without altering expression levels of total RhoA protein. This result suggests that RhoA may be in the downstream signaling pathway modulated by LIS1 dose. Our rescue experiments with CA-RhoA also suggest that GTPase activity and biological functions of RhoA may be indirectly modulated by LIS1 expression levels in the mitotic cells. These arguments certainly do not rule out a low level of direct interaction between LIS1 and Anilin/RhoA, which would require GST-pull down assays or two-hybrid phage array with recombinant proteins. We believe these are beyond the scope of the current manuscript and can be performed in the future.

We have inserted text in the last paragraph of the subsection “Mislocalization of RhoA and contractile ring components in mitosis of *Pafah1b1* mutant MEFs".

2) Quantitative data need to be improved. Please indicate the exact sample size of experiment whenever possible. Very often the sample sizes in the manuscript are shown as "more than twenty cells", "more than twenty apical NPCs", "at least thirty cells". The authors should indicate the n number in each experiment and figure and present the data as "mean ± S.E.M" as much as possible. The population of abnormal division of NPCs were not quantified in Figure 1 and Figure 2.

We thank all reviewers for pointing out the importance of providing accurate quantification and sample number/size. We made a new source data files to include all this data (n=brain samples or MEF cell number, N=independently replicated experiments). All data associated with the each figure were represented with detailed description. The subgrouping category identifying different groups based on the cellular phenotypes was also explained (Figures 1, 2, 3, 6, 8, 11). The graphs in Figure 4 and Figure 5 have mean ± S.E.M. values (also integrated in the texts in the Results section) and the exact *p*-values from Student’s two-tailed *t*-test were provided in source data files. We also quantified the phenotypes of RG cells included in Figure 1 and Figure 2 (Figure 1—source data 1, Figure 2—source data 1).

As noted above, we have included new source data files. Text in Materials and methods, subsection “Sample size, Replicates, and Statistical analysis”, was updated.

3) It would be helpful to connect the disordered division, and shifts in asymmetric and symmetric division that the authors report to the ultimate functional impact in the neuronal cell population. Further could the authors clarify if this dependence is somehow specific to this subtype of neuronal precursors?

We thank the reviewers for raising these important questions regarding the functional outcomes affecting neuronal subtypes born from *Pafah1b1*-deficiency in NPCs and the long-term effects of daughter cell fates. At around E14.5, asymmetric divisions generate one NPC and one neuron and, in contrast, symmetric divisions generate two NPCs. Since *Pafah1b1*-deficiency in NPCs induces an increase in asymmetric divisions with oblique spindle angles, it is expected that number of neuronal cells may be ultimately increased. In accordance with this, *Pafah1b1*-deficient neocortex had an increase in TUJ1-positive cells abnormally located in the VZ, indicating delayed migration of post-mitotic neurons (Gambello et al., 2003). Previously published data from our laboratory of cleaved caspase-3 staining and TUNEL assay from *Pafah1b1*-deficient neocortex (Yingling et al., 2008) indicate that the apoptotic events were increased at E14.5. This suggests that final cell fate determination affecting a balance between cell death and survival were disrupted by *Pafah1b1*-deficiency in NPCs.

In our current study, we mainly analyzed RG phenotypes from various *Pafah1b1* mutants at E14.5 mid-gestational time point. We agreed with the reviewers that it is important to assess the long-term effects from LIS1 dose changes in RGs by the follow-up of later-gestational time point. In our previous study, we demonstrated that there were LIS1 dose-dependent changes in cortical patterning in the neocortex. Although *Pafah1b1*-deficient neocortex (*Pafah1b1 hc/ko*) at E15.5 had a distinct cortical plate, due to weak formation of the subplate boundary, the cortical plate (CP) was partly diffused to the intermediate zone. At a later time point P0 (post-natal day 0), the diffuseness and thinning of E15.5-born BrdU labeled cells were evident in *Pafah1b1* mutant neocortex with reduced cortical thickness (Gambello et al., 2003). Consistent with conventional *Pafah1b1*-deficient mutant (*Pafah1b1 hc/ko),* NPC-specific *Pafah1b1* CKO mice (*GFAP-Cre; Pafah1b1 hc/hc*) also exhibited similar cortical patterning defects at P5. Immunohistochemistry analyses with upper layer-specific FOXP1 (Layer III/VI) and CUX1 (Layer II/III) projection neuronal makers suggest that *Pafah1b1* CKO neocortex has significantly reduced immunoreactive cells in both layers compared with control (*GFAP-Cre; Pafah1b1 hc/+)* (Youn et al., 2009). Altogether, increased TUJ1+ cells and decreased FOXP1+ and CUX1+ cells in *Pafah1b1*-deficient neocortex indicate that RG-derived neuronal populations have impairments in neuronal subtype specification and total neurogenesis inputs. Reduced LIS1 dose in apical RGs has long-range adverse impact on CP patterning and layering by dysregulating neuronal production and subtype alterations of the daughter cells.

To address these points in the manuscript, we have added text to the seventh paragraph of the Discussion.

4) How much of the data generated in the MEFs is relevant for the NPCs of the neocortex? Is it possible to isolate neuronal precursors from the Lis1 null mice, or differentiate neurons towards these lineage to validate that some of the findings in MEFs hold true for NPCs? For example, is it possible to check whether the separation defects so effectively demonstrated in MEFs actually also occur in NPCs?

Effects of LIS1 on long-term functional outcomes and neuronal subtype/lineage specification in neocortical development were discussed in the response to the point 3 above. Additional text was added to the Discussion to address these questions. Points discussed in this point 4 and point 5 were addressed together below.

5) The paper risks becoming somewhat descriptive at times. Better integration of the findings especially those in the MEFs, and connecting them to how Lis1 may impact NPC generation and functionality could help resolve this.

[Points 4 and 5]: MEFs and NPCs.

We agree with the reviewers about importance of additional explanations of descriptions covering key limitations and advantages of MEFs compared with the usage of NPCs. We initially performed in vitro NPC primary culture from WT (*Pafah1b1 +/+)* and *Pahah1b1 hc/ko* mice. Conventional NPC protocols of NPC utilizing basal N2 medium with incubation on coated plates was used (Youn et al., 2013 – previously described in our group’s publication). Although WT NPCs normally grew in this culture condition, *Pafah1b1 hc/ko* NPCs did not, probably due to severe mitotic arrest at prometaphase and metaphase. It was not possible to maintain and propagate *Pafah1b1 hc/ko*NPCs in vitro. Therefore, we do not know whether *Pafah1b1*-deficient NPCs in vitro have exactly same phenotypes or even worse cytokinetic progression than MEFs in vitro. Our previously published study (Yingling et al., 2008) suggests that *Pafah1b1*-deficient NPCs have less proliferation/self-renewal capability with reduced phospho-Histone H3 mitotic index. It was assumed that asymmetric divisions generating neuronal cell types may increase in *Pafah1b1*-deficient NPCs due to alterations of vertical mitotic spindle angles.

Although *Pafah1b1-*deficient NPCs did not grow well in culture, we were able to successfully propagate *Pafah1b1 hc/ko* MEFs (derived from E14.5) and perform several in-depth cellular analyses by immunocytochemistry and live-cell imaging that were not technically possible in *Pafah1b1*-deficient NPCs. This suggests that *Pafah1b1*-deficiency in different cell types dissected from different tissue (CNS vs. skin) leads to distinct abilities to grow in vitro. This is not surprising, since humans with heterozygous loss of LIS1 display specific brain phenotypes without noticeable defects in other tissues, Mouse neocortical NPCs express specific nuclear transcription factors (such as PAX6, SOX2) to maintain its proliferative capacity and potency to promote neurogenesis. MEFs are subsets of the fibroblasts eventually undergo cellular senescence. NPCs uniquely undergo complex basal-to-apical abcission-mediated cytokinetic process in vivo. NPCs also generate one daughter cell with a different fate, a neuron with different lineage depending on the birth date and subtypes of NPCs in regionally specified neocortical domain. By contrast, MEFs only generate two MEFs with symmetric divisions.

However, there are many similarities which we believe allow us to use MEFs to study these phenotypes. Although MEFs and NPCs are isolated from the different source of tissue origins, LIS1 mutant cells expressed the same amount of LIS1 protein (35%) compared with WT (100%). LIS1-dose sensitivity in NPCs may lead to more severe mitotic phenotypes. Despite this disparity, both MEFs and NPCs have shared cellular and cytoarchitectural features during cytokinesis. We confirmed that NPCs retain apical RhoA and Anillin localization when they undergo cytokinesis at the ventricular surface where F-actin polarization occurred in vivo. RhoA and Anillin play crucial roles in determining cytokinetic furrow localization by cross-interacting with F-actin and Myosin II at the equatorial cell membrane and midbody. Thus, it appears that RhoA and Anillin-mediated cytokinetic furrow formation is controlled in a similar manner in both MEFs in vitro and apical NPCs in vivo (vRGs).

It is noteworthy that multiple in vitro studies have highlighted that genetic mutant MEFs are versatile and reliable cell types to investigate cellular mechanisms underlying cytokinetic defects, including cytokinesis defects in the brain (Janisch et al., 2013; Rosario et al., 2010; Cook et al., 2011; Serres et al., 2012). In genetically targeted MEFs with reduced protein dose (e.g. Kif20b, Plk4, Ect2, and P27), cytokinesis-associated phenotypes are found, including multi-nucleation and centrosome amplification. Dysregulated or mislocalized RhoA are commonly found as upstream mechanisms. Ectopic actomyosin localization away from the midbody leads to abcission failure and formation of aberrant inter-cellular bridge. Although transformed cell lines are widely used cell types to study cytokinesis-related mechanisms, primary cell types like MEFs derived from the well-defined genotypes are important tools to study targeting developmentally critical early embryonic time point. In particular, *Kif20b -/-* KO mutant MEFs exhibited no detectable AuroraB-positive midbody indicating failure in abcission during cytokinesis. Interestingly, *Kif20b -/-* RGs in the neocortex also had reduced numbers of AuroraB-positive midbodies at the apical surface of VZ (Janisch et al., 2013). Therefore, unlike RGs in the neocortex in vivo, the feasibility and durability of long-term live cell imaging of primary MEFs enabled us to measure variables and help reach better statistical power. Our work tracing the fluorescently labeled intracellular and membrane markers and immunocytochemistry with cytokinetic markers in *Pafah1b1* mutant MEFs are advantageous to elucidate the exact signaling pathways to regulate LIS1-dependent cytokinesis.

To address these points in the manuscript, we have added text to the second paragraph of the Discussion.

6) While the results are well documented, the text needs considerable improvement. The authors did raise interesting questions in the Introduction, for example: why do LIS1 mutants specifically display a decrease in neuroepithelial stem cells in the neocortex compared with a less catastrophic phenotype seen in radial glial (RG) progenitors?

A) We previously demonstrated that *Pafah1b1* deficiency in neural epithelial stem cells (NESCs) (E9.5~11.5) display more severe mitotic defects than radial glial progenitor cells (RGPCs) (E12.5~E14.5) during mouse neocortical development(Yingling et al., 2008). LIS1 is essential for precise control of mitotic spindle orientation in both NESCs and RGPCs. Controlled gene deletion of *Pafha1b1* in vivo in neuroepithelial stem cells, where cleavage is uniformly vertical and symmetrical, provokes rapid apoptosis of those cells, while radial glial progenitors are less affected because their divisions are symmetric and asymmetric.In the present study, we designed our experiments to dissect the mouse neocortices from E14.5 time-point. We selected these time-points because we narrowed the developing critical window of active and rapid neurogenesis in order to examine the effects of *Pafah1b1*-deficiency on NPC daughter cell’s fate and cytokinetic separation. Therefore, almost all apical NPCs observed in the current study are expected to be RGPCs. We aimed to get mild but somewhat measurable defects in cytokinesis of RGPCs which overcome significantly increased cell death events and severe mitotic arrest seen NESCs.

To address these points in the manuscript, we have added text to the eighth paragraph of the Discussion.

Does LIS1 have differential roles in mitotic division time between ventricular and subventricular radial glial progenitors? However, there is little answer to these questions in the paper with the current experimental design and methods. The work was partly carried out on mouse progenitor cells without differentiation of epithelial stem cell and radial glia cells.

B) In human organoid models, the time between rounding of cells and their separation is similar between control vRGs and oRGs. However, in MDS organoids, there was a profound delay in oRG division and vRG division (Bershteyn et al., 2017), a point we discussed in the text. In the mouse neocortex, there are very few oRGs since this progenitor type is greatly expanded in primates. Mouse RGPCs are nearly all ventricular RGs (vRGs) associated physically with apical membrane of VZ surface, not outer RGs (oRGs) located at the subventicular zone (SVZ) area.

To address these points in the manuscript, we have added text to the last paragraph of the Discussion.

In response to these questions, additional information discussing the differences between oRG and vRG and human vs. mouse neocortical development were added in the first paragraph of the subsection “Mislocalization of RhoA and Anillin in *Pafah1b1* mutant neocortical neural progenitor cells (NPCs)” (a similar issue was covered in (D) section).

…and partly on MEFs which have significant limitations relevant to answer the main the questions. The difference between neuronal migration and cortical development in…

C) We thank the reviewers for mentioning this crucial point highlighting the difference and limitation of simply applying the findings of MEF live-cell imaging analyses to NPCs in the developing neocortex. These concerns were addressed in the revised manuscript, as we discussed in Points (D) and (F).

To address these points in the manuscript, we have added text to the second paragraph of the Discussion.

Lisencephalic rodent (mouse) and gyrensephlaic primate (human) is not sufficiently emphasized. The Introduction and Discussion are not scholarly written and historically informed to be useful to a multidisciplinary audience of a journal like eLife.

D) A new paragraph was integrated to respond to this point (Discussion, last paragraph).

For example, while citing themselves profusely (10 times!?)

E) Yingling et al., 2008 manuscript was reduced to ‘7’ citations in the current text. This particular citation has most relevant previous findings related to LIS1 dose-dependent RGPC phenotypes and mitotic spindle regulation. Therefore, we still kept this reference in multiple locations of the middle of the revised manuscript where citation is still critical and necessary to make understood the readers.

To address these points in the manuscript, we have added text in the Introduction, Results and Discussion sections.

The seminal papers by Reiner Orly on LIS1 gene are conspicuously missing.

F) We are sorry to miss certain historically important citations in this text. We did not omit it intentionally from the initial manuscript. When we followed the guideline of the reference numbers in *eLife* submission, various citations were subtracted. The Reiner et al., 1993 citation was added in the first paragraph of the Introduction, and in theReferences section.